# Systematic bacterialization of yeast genes identifies a near-universally swappable pathway

Aashiq H Kachroo[1*†‡], Jon M Laurent[1†§], Azat Akhmetov[1], Madelyn Szilagyi-Jones[1], Claire D McWhite[1], Alice Zhao[1], Edward M Marcotte[1,2*]

[1]Center for Systems and Synthetic Biology, Institute for Cellular and Molecular Biology, University of Texas at Austin, Austin, United States; [2]Department of Molecular Biosciences, University of Texas at Austin, Austin, United States

*For correspondence: aashiq. kachroo@concordia.ca (AHK); marcotte@icmb.utexas.edu (EMM)

[†]These authors contributed equally to this work

Present address: [‡]Centre for Applied Synthetic Biology, Department of Biology, Concordia University, Montreal, QC, Canada; [§]Institute for Systems Genetics, New York University Langone Medical Center, New York, NY, United States

Competing interests: The authors declare that no competing interests exist.

**Abstract** Eukaryotes and prokaryotes last shared a common ancestor ~2 billion years ago, and while many present-day genes in these lineages predate this divergence, the extent to which these genes still perform their ancestral functions is largely unknown. To test principles governing retention of ancient function, we asked if prokaryotic genes could replace their essential eukaryotic orthologs. We systematically replaced essential genes in yeast by their 1:1 orthologs from *Escherichia coli*. After accounting for mitochondrial localization and alternative start codons, 31 out of 51 bacterial genes tested (61%) could complement a lethal growth defect and replace their yeast orthologs with minimal effects on growth rate. Replaceability was determined on a pathway-by-pathway basis; codon usage, abundance, and sequence similarity contributed predictive power. The heme biosynthesis pathway was particularly amenable to inter-kingdom exchange, with each yeast enzyme replaceable by its bacterial, human, or plant ortholog, suggesting it as a near-universally swappable pathway.

## Introduction

Despite over 2 billion years of divergence, eukaryotes and prokaryotes still share hundreds of genes (*Theobald, 2010*; *O'Brien et al., 2005*; *Brown and Doolittle, 1997*; *Martin and Müller, 1998*). Though these ancient genes are identifiable as orthologs at the sequence level, the preservation of original protein function across such deep timescales has not been systematically explored. The function of certain genes could potentially become frozen in place in the course of evolution, sheltered from lineage-specific functional alterations introduced by mutations, gene fusions, and non-orthologous gene displacements. Such functionally frozen genes would in principle be able to substitute for their least-diverged ortholog in any other species. Searching for such gene replaceability between species thus serves to test a core assumption of the ortholog-function conjecture: that orthologs retain ancestral function (*Gabaldón and Koonin, 2013*). This conjecture forms the basis of most modern biomedical research and is widely used to predict new gene function across organisms (*Lee et al., 2007*).

There are many individual examples of genes from one species functioning for their orthologous counterparts in a different species (*Cherry et al., 2012*; *Heinicke et al., 2007*), but this trend has only recently begun to be explored systematically, with several large-scale studies substituting human genes for yeast genes and confirming that many human orthologs can successfully replace their yeast counterparts (*Kachroo et al., 2015*; *Sun et al., 2016*; *Hamza et al., 2015*). At the level of evolutionary divergence of yeast and humans, such data demonstrate widespread functional conservation, even after 1 billion years of divergence. The ability of human genes to functionally replace

**eLife digest** All life on Earth – from bacteria to human beings – can be traced back to a common ancestor that lived over three billion years ago. As a result, modern-day organisms share many essential parts of life's molecular machinery, such as certain genes and proteins. Yet there are also vital differences that allow scientists to divide almost all living things into one of two groups, known as prokaryotes and eukaryotes. Prokaryotes are all simple, single-celled organisms, such as bacteria; while eukaryotes include more complex organisms, such as plants, animals and fungi.

Scientists have previously found that eukaryotes and prokaryotes have hundreds of genes in common, even though they have evolved separately for over two billion years. As different species evolve, however, their genes mutate and change, potentially affecting the way they work. So, although scientists can recognize equivalent genes between species, they are not sure if they work the same way as they did in the species' ancient ancestors.

To investigate this, one-by-one Kachroo, Laurent et al. replaced over 50 genes in baker's yeast (a eukaryote) with their equivalent gene from *E. coli* bacteria (a prokaryote). If the yeast cells grow healthily after the gene is replaced, it means that that gene works in a similar way in both bacteria and yeast. That, in turn, suggests it is likely that the genes work as they did in the last common ancestor of bacteria and yeast.

The experiments found that most of the tested *E. coli* genes (61% to be precise) could successfully replace equivalent genes in yeast cells. Moreover, genes often work together in groups, and Kachroo, Laurent et al. found that genes in some groups were more successfully replaced than others. For example, nearly every gene that is important for producing a molecule called heme could be freely swapped from bacteria, plants and humans into yeast. This group of genes has probably worked the same way in different species for billions of years.

Understanding why genes sometimes change how they work is an important question for scientists studying evolution, but this knowledge has other uses. For example, people need heme to, amongst other things, carry oxygen in their blood, and a mutation in a gene in the heme production pathway causes a disease called porphyria. Scientists could replace genes in yeast cells to better model the disease in humans, leading to a better understanding of its causes and more efficient development of new drugs.

their yeast orthologs is not strongly predicted by the similarity of sequences, but rather at the level of specific pathways or processes, wherein all genes in a process or pathway tend to be similarly replaceable, or not (*Kachroo et al., 2015*).

However, in the timescale of evolution, yeast and humans are relatively similar – both eukaryotes that share thousands of genes and the majority of their core biological processes. Data on eukaryote – prokaryote functional gene replacement are sparse (*Heinicke et al., 2007*). These cross-domain replacements represent a maximum test of the ability of genes to retain their ancestral function across time. Eukaryotic and bacterial genes have been, for the most part, evolving independently since at least the archaeal ancestor of eukaryotes endosymbiotically acquired its bacterial mitochondrion. In eukaryotes, the function of these genes would have had to survive the development of vastly different genome structures, cell division modalities, cell wall compositions, and subcellular compartmentalizations which occurred during eukaryogenesis. Prokaryotic and eukaryotic orthologs also diverged significantly at the amino acid sequence level (*O'Brien et al., 2005*) and evolved distinct expression patterns and codon usages (*Sharp et al., 1993*; *Bulmer, 1991*). Nonetheless, eukaryotes and bacteria are known to use many of the same orthologs to perform the same metabolic enzymatic reactions (*Jardine et al., 2002*; *Peregrin-Alvarez et al., 2003*).

Thus, in order to more systematically determine the replaceability of orthologs across such deep timescales, we asked in this study how many conserved *E. coli* genes can successfully substitute for their yeast orthologs. We focused on those genes that are essential for viability in yeast, allowing us to assay for the complementation of otherwise lethal growth defects. We analysed many features of the proteins and ortholog pairs to identify which properties best explained replaceability, finding that replaceability was often determined at the level of specific pathways and processes, with all

genes in a pathway or process similarly replaceable. Start codon choice and eukaryote-specific sub-cellular localization were also critical determinants of replaceability. We discovered that certain core biological processes have remained largely unchanged since the last common ancestor of bacteria, yeast, and humans. In particular, heme biosynthesis pathway enzymes appear to be generally exchangeable between prokaryotic and eukaryotic organisms, broadly retaining ancestral functions across the tree of life over 2 billion years of independent evolution, even when accompanied by evolved changes in enzyme subcellular localization.

## Results and discussion

### Many *E. coli* genes successfully complement lethal defects in their yeast orthologs

We focused our efforts on the set of genes with 1:1 orthology between *E. coli* and yeast and that are known to be essential for yeast growth in standard laboratory conditions (*Figure 1A*). Each *E. coli* open reading frame (ORF) was cloned into a single-copy yeast centromeric (CEN) plasmid under the transcriptional control of a constitutive GPD promoter. Complementation assays were carried out using two types of conditionally essential yeast alleles, consisting of temperature-sensitive (TS) haploid and heterozygous diploid deletion strains. In the case of the heterozygous diploid deletion strains, the respective yeast gene null allele could be genetically segregated via sporulation, allowing selection for haploid yeast with the null allele (selected for in the presence of the antibiotic G418) or the wild-type yeast gene (in the absence of G418) (*Figure 1B*, Top panel). In the case of TS haploid yeast strains, the temperature sensitive yeast proteins functioned normally at the permissive temperature (25°C) but could be conditionally inactivated at the non-permissive temperature (36°C) in order to test for gene replaceability (*Figure 1B*, Bottom panel). Overall, we could perform informative complementation assays for 51 of the 58 orthologs, as shown for the examples in *Figure 1B*.

Of the 51 *E. coli* genes tested, 25 successfully complemented lethal growth defects in the corresponding yeast strains (*Figure 1—figure supplement 1A,B and C*; *Supplementary file 1*). In nearly all cases, despite plasmid-based expression of the complementing genes, the bacterialized strains grew comparably to the parental, wild type yeast strain, in both synthetic defined medium (SD - Ura + G418) (*Figure 1C*) and rich medium (YPD + G418) (*Figure 1—figure supplement 1D*). We further verified complementation specificity by testing for plasmid loss (see Materials and methods and (*Supplementary file 1*) and sequence verifying all clones. We have previously demonstrated that plasmid-borne copies of yeast genes complemented their corresponding heterozygous diploid deletion alleles at a high rate (100% for 29 strains tested in *Kachroo et al., 2015*), but as an additional control, we repeated this test for six yeast strains where the *E. coli* gene failed to rescue, confirming that the corresponding yeast genes were able to complement the growth defect when expressed on a CEN plasmid under the control of the constitutive GPD promoter (*Figure 1—figure supplement 2* and *Sc-HEM1* as reported in *Figure 4—figure supplement 1*).

### Mitochondrial localization and start codon choice both affect replaceability

Many eukaryotic orthologs of prokaryotic genes function in specific subcellular compartments absent from prokaryotes, and consistent with this trend, 15 of the 51 tested *E. coli* genes have mitochondrially-localized yeast orthologs (*Cherry et al., 2012*). Because all but one of these 15 genes were unable to replace their yeast ortholog, we reasoned that lack of mitochondria targeting might account for their failed complementation. We added the mitochondrial localization signal (MLS) from the yeast *MIP1* gene to each of the 14 non-replaceable *E. coli* genes and repeated the complementation assays. Four genes could now functionally replace their yeast equivalents (*Figure 2A*, *Figure 2—figure supplement 1*), restoring growth rates to be nearly or fully comparable with the parental strain (*Figure 2B*). We verified mitochondrial localization by fusing the *E. coli* Ec-MLS-HscB and Ec-MLS-IlvD proteins with enhanced green fluorescent protein (EGFP) and confirming correct trafficking of the EGFP-tagged proteins to yeast mitochondria (*Figure 2C*).

Bacterial genes also occasionally lack a standard ATG start codon, with ~14% of all *E. coli* ORFs employing an alternative start codon (*Blattner et al., 1997*). Three of the tested non-replaceable *E. coli* genes used a GTG start codon while one used ATT. We therefore used site-directed

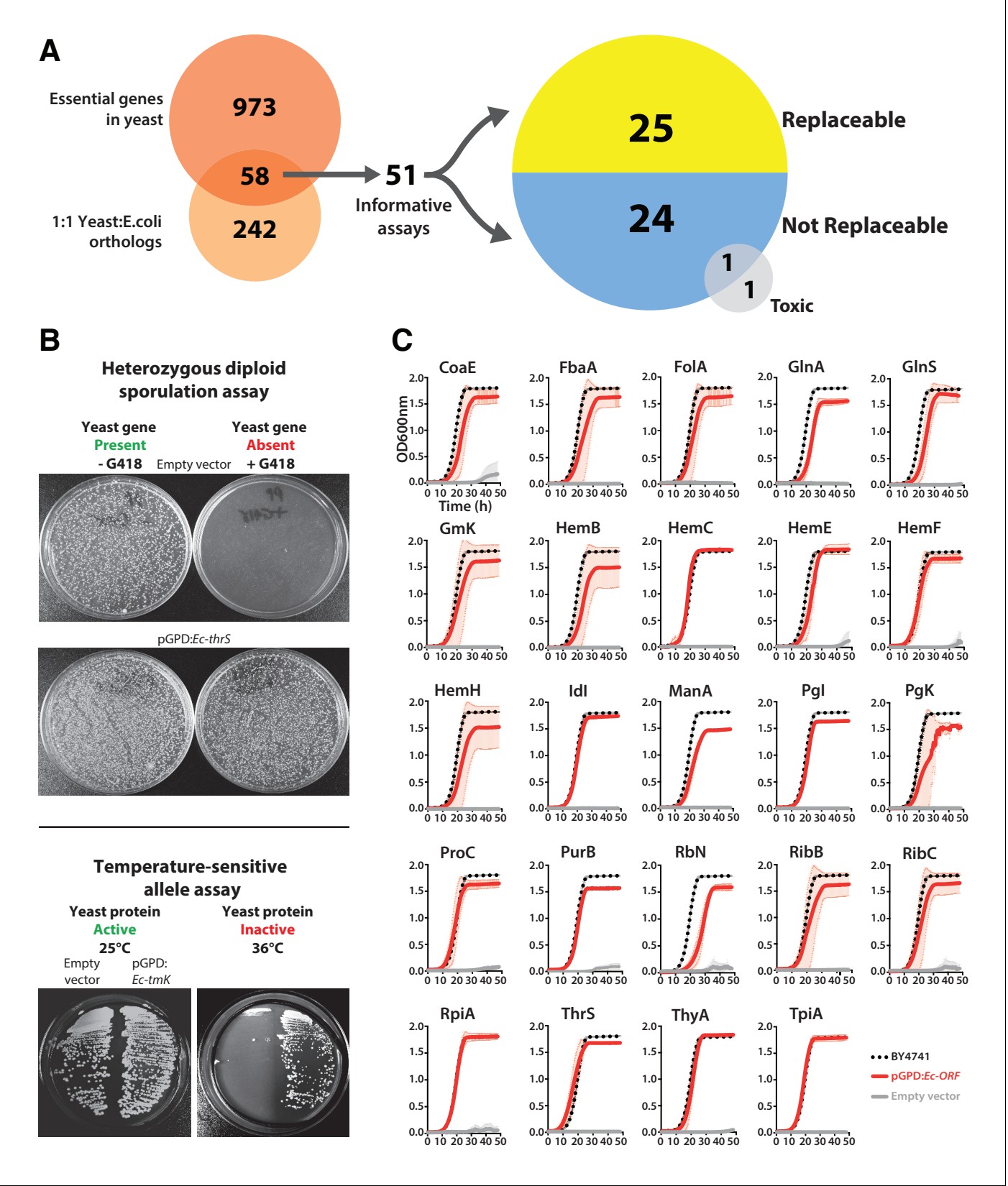

**Figure 1.** Many *E. coli* genes efficiently complement lethal growth defects in their yeast counterparts. (**A**) Yeast and *E. coli* share hundreds of genes, 58 of which are essential in yeast and have clear 1:1 orthologs in either species. *E. coli* genes were cloned into a yeast expression vector under the control

*Figure 1 continued on next page*

*Figure 1 continued*

of a GPD promoter. 51 of these 58 *E. coli* genes provided informative assays for replaceability in yeast. Initial results from these complementation assays revealed that 25 of 51 (~49%) *E. coli* genes could functionally replace their orthologous yeast counterparts. (B) Complementation assays were performed in two different yeast strain backgrounds, as shown for representative assays. In the case of a yeast strain with a temperature-sensitive allele of the yeast gene *Sc-cdc8*, cells carrying the empty vector control grow at the permissive-temperature (25°C, yeast protein active) but not the restrictive-temperature (36°C, yeast protein inactive), unlike cells expressing the *E. coli* ortholog (*Ec-tmK*), indicating that the *E. coli* gene can functionally replace the yeast gene. In the case of yeast heterozygous diploid (*Sc-ths1Δ/Sc-THS1*) deletion strain, cells are sporulated and haploid progeny grown on selective medium (-Ura -Arg -His -Leu + Can) in the absence (yeast gene present) or presence of G418 (200 µg/ml) (yeast gene absent). Cells expressing the *E. coli* ortholog (*Ec-thrS*) grow on G418-containing medium, unlike cells carrying the empty vector control, indicating successful complementation. (C) Haploid yeast gene deletion strains carrying plasmids expressing functionally replacing *E. coli* genes (red solid-lines) generally exhibit comparable growth rates to the wild type parental yeast strain BY4741 (black dotted-lines). The empty vector control (grey solid-line) showed no such growth rescue in the presence of G418. Mean and standard deviation plotted with N = 3.

The following figure supplements are available for figure 1:

**Figure supplement 1.** Complementation assays performed in a 96-well format in two different yeast strain backgrounds (*Supplementary file 1*).

**Figure supplement 2.** Constitutive plasmid expression of yeast genes efficiently replaced the corresponding genomic copies for 6 non-replaceable alleles.

---

mutagenesis to introduce canonical ATG start codons, then re-assayed for complementation. After changing their start codons to ATG, two of these four *E. coli* genes, *Ec-rcsC* and *Ec-tadA*, could now replace their yeast orthologs (*Figure 2B*).

Overall, after accounting for mitochondrial localization and alternative start codons and combining results from all assays, a total of 31 out of 51 tested *E. coli* genes could successfully replace their essential yeast orthologs (*Figure 2*). Thus, in a majority (61%) of our tests, both the current day prokaryotic and eukaryotic proteins must have retained their critical ancestral functions such that the prokaryotic proteins could carry out the essential roles of their eukaryotic orthologs well enough to support yeast cell growth. In one-fifth of the cases, replaceability depended on proper subcellular localization or start codon choice to express the prokaryotic gene in the proper eukaryotic context.

## Replaceability varies strongly across different biological processes

Given that we observed both replaceable and non-replaceable genes, we sought to determine properties of the tested genes that best explained successful replacements. We considered 22 features of the tested genes, including protein lengths, interactions, sequence similarities, codon usages, and expression levels. We calculated the predictive utility of each feature as the area under a Receiver Operating Characteristic curve (AUC) (*Figure 3A*; *Supplementary file 2*). Notably, the extent of protein sequence similarity between orthologs was not a highly predictive feature. A large portion of the tested *E. coli* and yeast orthologs showed only 20–30% identical amino acid sequences and roughly half of these genes were replaceable; in contrast, the three most divergent orthologs replaced, each showing less than 20% identity (*Figure 3B*). As we observed a non-monotonic relationship between sequence identity and replaceability, potentially explained by replaceability differences among different functional categories of genes, we tested for the enrichment of particular GO Biological Process (defined by Gene Ontology Slim annotations (*Ashburner et al., 2000*) or KEGG categories (*Kanehisa and Goto, 2000*) within the individual bins of sequence identity in *Figure 3B*. Aside from an enrichment in glucose metabolism genes (3 of the 7) in the 40–50% identity range, we did not find evidence for strong pathway-specific biases that would explain the observed relationship between sequence identity and replaceability. We did observe moderate predictive power for some measures of codon bias, especially those related to codon optimality within *E. coli*, and less so for codon optimality within a yeast context; more highly optimized *E. coli* codon usage correlated with a lower replaceability rate.

Instead, the strongest predictive features related to specific pathways and processes, much as we and others have observed for successful humanization of yeast (*Kachroo et al., 2015*; *Sun et al., 2016*; *Hamza et al., 2015*). This trend was most evident in the observation that a gene was more likely to replace (or not) if it had a higher fraction of interaction partners that also replaced (or not).

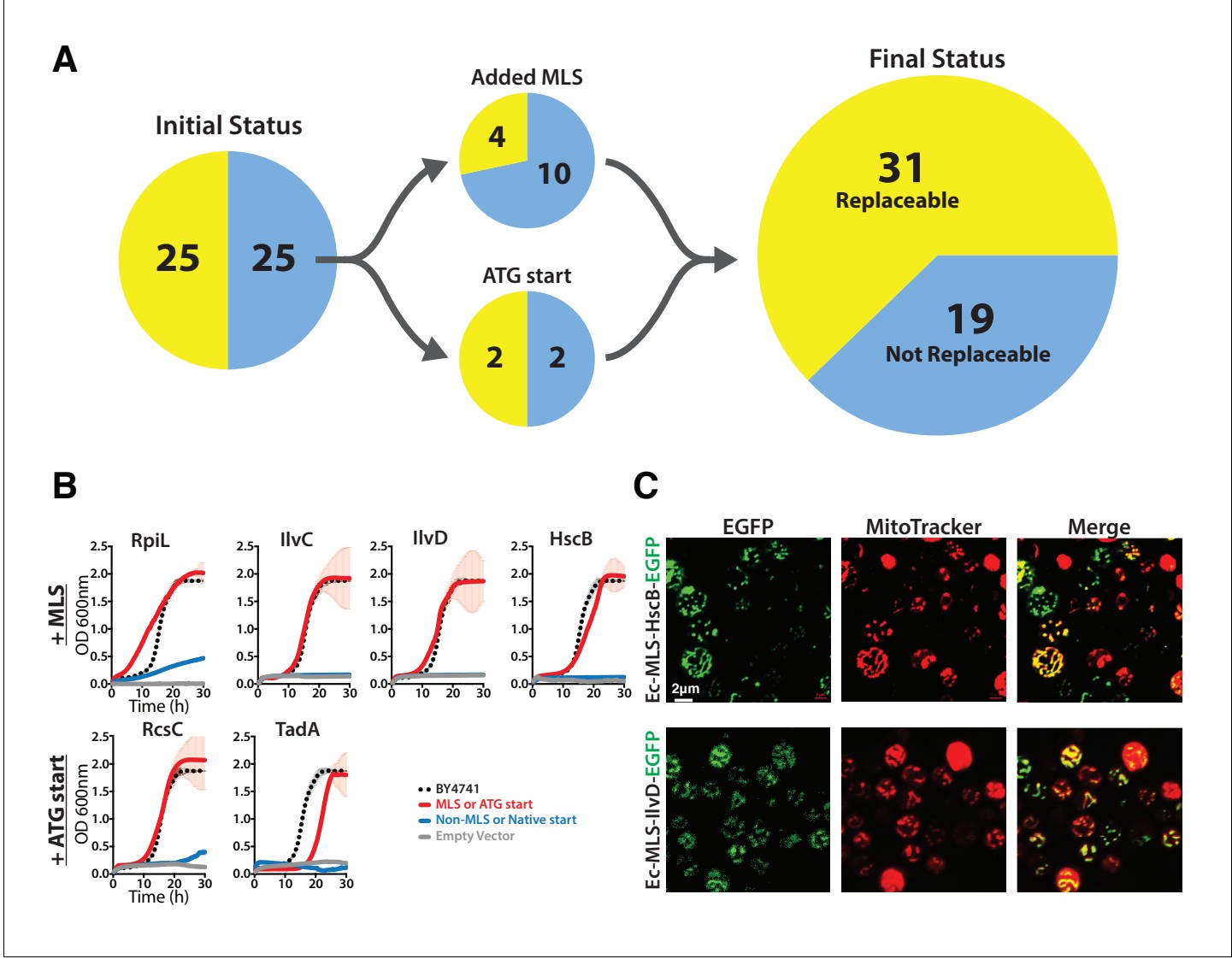

**Figure 2.** The addition of a mitochondrial localization signal (MLS) and mutation of start codons from GTG to ATG allows some *E. coli* genes to swap for their respective yeast orthologs. (**A**) 14 of the 25 non-replaceable *E. coli* genes were predicted to function in mitochondria in yeast. 4 of 14 were replaceable after adding the MLS at the N-termini of the *E. coli* genes. Site-specific mutagenesis of *E. coli* gene start codon from GTG to ATG allowed two to functionally complement the corresponding yeast genes bringing the total number *E. coli* genes that functionally replace yeast genes to 31 of 51 (~61%). (**B**) Haploid yeast gene deletion strains carrying mitochondrially localized *E. coli* genes rescued the growth defect of the yeast gene (red solid-line) comparable to the wild type yeast (black dashed-line). The empty vector control (grey solid-line) and the yeast cells expressing of *E. coli* gene without MLS (blue-solid line) showed no such growth rescue in the presence of G418. Mean and standard deviation plotted with N = 3. (**C**) EGFP-tagged *E. coli* genes that functionally replaced the yeast gene function were imaged after MitoTracker red staining. EGFP-tagged Ec-MLS-HscB and Ec-MLS-IlvD (green) show colocalization with MitoTracker red stained mitochondria (red).

The following figure supplement is available for figure 2:

**Figure supplement 1.** Some *E. coli* genes require a yeast mitochondrial localization signal to efficiently replace.

Consequently, different biological processes (as defined by GO) displayed varied replaceability, with metabolic processes being largely replaceable, while processes known to be divergent, including ribosomal processing, were much less replaceable (*Figure 3C*). This trend suggests an explanation for why optimized *E. coli* codons predicted worse replaceability, as *E. coli* genes with optimized codons predominantly tend to be highly expressed ribosomal and translational proteins

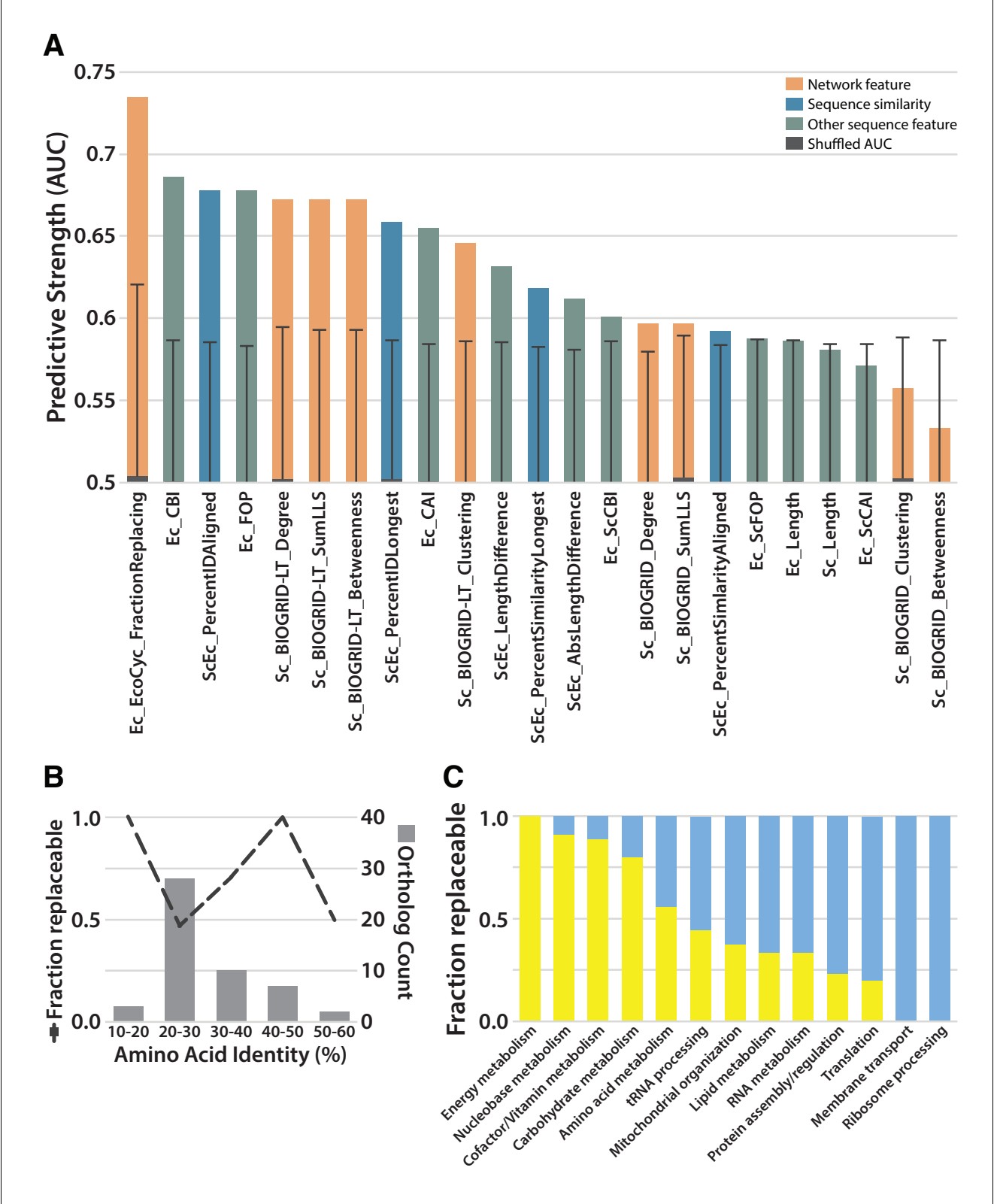

**Figure 3.** Replaceability of *E. coli* genes is a modular phenomenon. (**A**) Several quantitative properties of the tested genes were assessed for their ability to predict replaceability, measured as the area under a receiver operating characteristic curve (AUC). Having a high fraction of interaction partners that replace was the most predictive property tested, suggesting that the ability to replace is a modular phenomenon whereby genes functioning together are similarly able to replace. A Random Forest classifier constructed with all attributes boosted the maximum AUC to 0.79. (**B**) As
*Figure 3 continued on next page*

*Figure 3 continued*

shown in (A), sequence similarity was not the most predictive feature. The fraction of replaceable genes in given ranges of similarity was variable, with the vast majority of orthologs being 20–30% identical, a range in which roughly half of proteins replaced. (C) Mapping of replaceability status onto yeast GO slim annotations revealed that GO categories have varying rates of replaceability, with core metabolic processes (e.g. energy metabolism, nucleobase metabolism) being largely replaceable while more specialized processes (e.g. protein assembly, membrane transport) were less so.

(*Saikia et al., 2016*). This is thus consistent with the notion that replaceability is determined at the level of the pathway or process, with codon choice and gene expression levels reflecting functional constraints of that process. Combining all of these features into a single predictor (after accounting for mitochondrial localization and alternative start codons), using a random forest classifier, improved our predictive power to a 0.79 AUC (*Figure 3A*), demonstrating that the features we investigated provide moderately orthogonal predictive information.

## Each yeast heme biosynthesis enzyme can be replaced by its *E. coli* equivalent, irrespective of orthology or localization

Nearly all the genes that we tested from the heme biosynthesis pathway were replaceable by their *E. coli* orthologs, which in combination with the evidence that replaceability was determined at the level of processes, led us to investigate the heme pathway in more depth. Most of the enzymatic reactions in the heme biosynthesis pathway are identical between *E. coli* and yeast, but there are clear differences in the way this pathway functions between the species (*Heinemann et al., 2008*). First, heme biosynthesis pathway precursors differ: Yeast condense succinyl-CoA and glycine to produce delta-aminolevulinate in a single enzymatic step catalyzed by Sc-Hem1, while *E. coli* produces delta-aminolevulinate in two steps using glutamyl-tRNA as a precursor (*Yin and Bauer, 2013*). Second, the bacterial heme pathway is largely cytosolic but in yeast it is partitioned between the mitochondria and cytosol (*Figure 4A*). We thus next considered these two key pathway differences in more detail. As a control, we first expressed the corresponding yeast genes on plasmids either under the control of constitutive GPD or the native yeast promoter (*Ho et al, 2009*) to test the effect of constitutive expression on functional replaceability. Except for *Sc-HEM4*, which showed toxicity when expressed constitutively, all the other yeast genes showed functional replaceability irrespective of the mode of expression (*Figure 4—figure supplement 1*).

In our initial screen, the *E. coli* ortholog of *Sc-HEM1*, *Ec-kbL*, failed to replace the yeast gene, an observation consistent with prior data showing that *Ec-kbL* does not take part in *E. coli* heme biosynthesis, but rather carries out an unrelated but mechanistically-similar oxido-reductase reaction involved in L-threonine degradation (*UniProt Consortium, 2015*; *Mukherjee and Dekker, 1990*). Instead, a two-step enzymatic reaction by *E. coli* proteins Ec-HemA and Ec-HemL produces the heme precursor, delta-aminolevulinate (*Schauer et al., 2002*; *Ilag and Jahn, 1992*). Since the initial steps of the pathway are localized to the mitochondria, we added the *Sc-MIP1* MLS to the 5' ends of these genes and expressed them simultaneously in the *Sc-HEM1* heterozygous diploid deletion strain. Co-expression of the two *E. coli* genes successfully replaced yeast gene function (*Figure 4—figure supplement 2A*). Additionally, two enzymes, Ec-HemD and Ec-HemG, were not identified as orthologs between *E. coli* and yeast, despite carrying out identical reactions to Sc-Hem4 and Sc-Hem14, respectively. Expression of these non-orthologous but functionally analogous *E. coli* genes in the respective yeast deletion strains showed that they were indeed able to successfully replace the yeast genes (*Figure 4—figure supplement 2B*). For these enzymes, the key determinants for successful replacement are thus their enzymatic reactions, rather than any other aspects of the genes.

Sc-Hem14 and Sc-Hem15 carry out the final two steps in yeast heme biosynthesis and are localized to the mitochondria (*Cherry et al., 2012*; *Koh et al, 2015*) (*Figure 4A*). Both genes were replaceable by the *E. coli* genes carrying out the analogous reactions, Ec-HemG (*Figure 4—figure supplement 2B*) and Ec-HemH (*Figure 1C*), despite the lack of targeting sequences for mitochondrial localization. As *E. coli* lack mitochondria, and Ec-HemG and Ec-HemH are both predicted to localize to the plasma membrane in *E. coli* (*Papanastasiou et al., 2013*), we thus assayed their localization in yeast when expressed as EGFP-fusion proteins. Strikingly, both localized to the yeast plasma membrane (*Figure 4—figure supplement 3*). In spite of failing to localize to the yeast

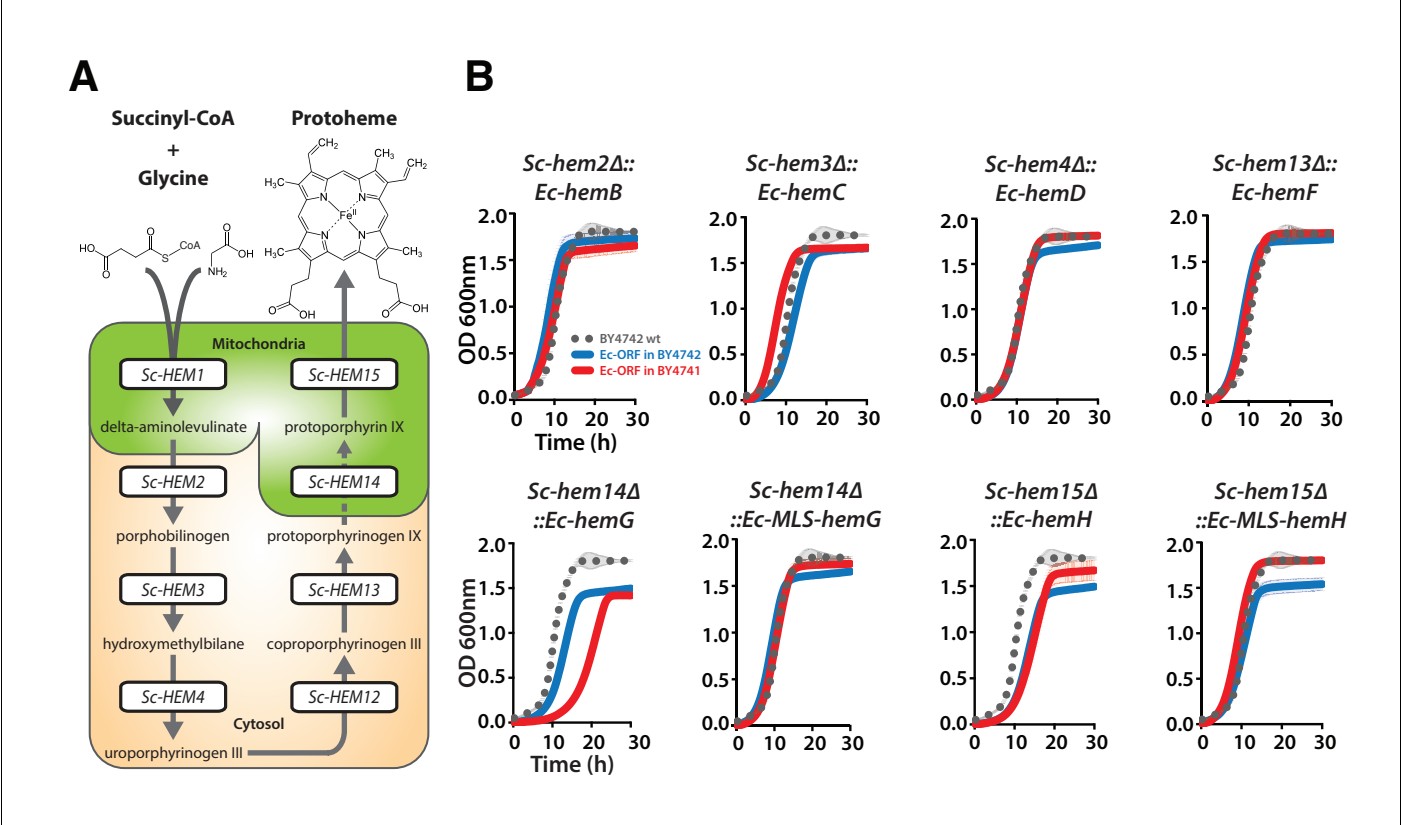

**Figure 4.** Bacterialization of yeast heme biosynthesis pathway genes at their native loci. (**A**) A schematic of the yeast heme pathway shows the beginning of the pathway in mitochondria using succinyl-CoA and glycine as precursors. The subsequent enzymatic reactions are cytosolic up until the penultimate and ultimate reactions which are mitochondrial. (**B**) Growth kinetics of CRISPR-Cas9 engineered yeast heme pathway genes replaced with the corresponding bacterial genes at their native yeast loci show efficient replaceability in both BY4741 (red solid-line) and BY4742 (blue solid-line) yeast strains. The wild type BY4741 growth curve is shown as a comparison (black dotted-line). Mean and standard deviation plotted with N = 3.

The following figure supplements are available for figure 4:

**Figure supplement 1.** Constitutive or native plasmid-based expression of the yeast heme biosynthesis genes generally efficiently complemented growth defects in the corresponding yeast gene deletion strains.

**Figure supplement 2.** Ec-hemA and Ec-hemL carry out the initial reaction in *E. coli* heme biosynthesis and are both required to complement Sc-HEM1 deletion in yeast, and non-orthologous yeast genes are replaced by *E. coli* genes that carry out the identical reaction.

**Figure supplement 3.** The penultimate and ultimate heme pathway enzymes in yeast are replaceable by their bacterial orthologs, in spite of mis-localizing to the plasma membrane.

**Figure supplement 4.** Confirmation of CRISPR-Cas9 mediated bacterialized yeast strains.

mitochondria, the bacterialized strains grew well compared to wild type yeast (*Figure 4—figure supplement 3*), suggesting that mitochondrial localization is not an absolute requirement for their functions, as many heme pathway intermediates are cytosolic. However, concurrent bacterialization of both yeast genes resulted in a viable but defective yeast strain (*Figure 4—figure supplement 2C*), suggesting that the fitness cost of mis-localizing both proteins is not tolerated well, potentially due to cumulative effects of reduced efficiency of the bacterial proteins, altered allosteric regulation in yeast, or the accumulation of heme precursors in the wrong compartment (cytosol) (*Yin and Bauer, 2013*).

Because heterologous expression using a constitutive promoter could be compensating for more subtle functional differences, we also wished to measure complementation after placing the bacterial orthologs under control of the native yeast gene regulation. We thus used CRISPR/Cas9-based precision genome engineering to genomically replace each of the heme biosynthesis pathway genes in turn in yeast (except Sc-*HEM12*) with its respective *E. coli* counterpart, from start to stop codon, while retaining the native promoters, terminators, and chromosomal context of the yeast genes (*Figure 4B*, *Figure 4—figure supplement 4*). All strains but two grew comparably to the wild-type; the *Sc-hem14Δ::Ec-hemG* and *Sc-hem15Δ::Ec-hemH* strains showed modest growth defects (*Figure 4B*). Because these two yeast proteins are known to be mitochondrially localized (*Cherry et al., 2012*), we re-engineered each of the *Ec-hemG* and *Ec-hemH* ORFs into the yeast chromosome such that each gene's native yeast MLS was retained (*Sc-hem14Δ::Ec-MLS-hemG* and *Sc-hem15Δ::Ec-MLS-hemH*). The addition of the yeast MLS to each *E. coli* ORF completely ameliorated growth defects from the ORFs alone (*Figure 4B*).

Thus, the yeast heme biosynthesis pathway appears entirely replaceable, one gene at a time, by their corresponding bacterial genes, whether expressed constitutively from plasmids or directly integrated into chromosomes under native yeast transcriptional regulation. The extent of replaceability strongly suggests that ancestral functions in these genes (with the obvious exception of the non-orthologous steps) have remained intact and unaltered, at least in terms of critical, enzymatic functionality. Mitochondrial localization of several of the enzymes, while needed to fully recover growth rates, is not essential for viability.

## Bacterialization with the *E. coli* ferrochelatase induces a yeast phenotype resembling human porphyria

*Ec-hemH* and *Sc-HEM15* encode ferrochelatase, the enzyme responsible for adding iron to the porphyrin ring of protoporphyrin IX to produce protoheme (*Figure 4A*). In the course of constructing the CRISPR-edited yeast strains, we noticed that the *Sc-hem15Δ::Ec-hemH* yeast strain turned pink on a standard YPD agar medium upon prolonged incubation of 3–4 days (*Figure 5A*). This phenotype was consistent across all independently obtained, sequence verified yeast clones. The pink phenotype decreased dramatically in the *Sc-hem15Δ::Ec-MLS-hemH* strains in which Ec-HemH was correctly localized to the mitochondria by addition of an MLS.

We speculated that the pink phenotype was likely due to aberrant accumulation of porphyrin intermediates, presumably leading to their secretion, as we observed that the pigment could be washed off the cells. Therefore, we chemically extracted the pink pigment from *Sc-hem15Δ::Ec-hemH*, *Sc-hem15Δ::Ec-MLS-hemH* and wild type yeast cells (Materials and methods) and performed fluorescence spectroscopy to determine that the pigment likely corresponds to protoporphyrin IX (*Figure 5B*, *Figure 5—figure supplement 1*).

In order to determine whether protein mis-localization contributed to the phenotype, we removed the MLS from the native yeast gene. Several clones of the *Sc-ΔMLS-HEM15* yeast strain displayed similar extracellular pigment (*Figure 5B*, *Figure 5—figure supplement 1*). These results suggest that mislocalized plasma membrane-bound Ec-HemH in yeast does not convert protoporphyrin IX to protoheme efficiently, resulting in the accumulation and secretion of protoporphyrin IX. We further tested this line of reasoning by deleting the gene for the preceding step in the pathway, Sc-*HEM14*, which encodes the enzyme protoporphyrinogen oxidase and is responsible for making protoporphyrin IX. Using CRISPR, we deleted the Sc-*HEM14* ORF in wild type BY4741, Sc-*hem15Δ*::Ec-HemH, and Sc-*hem15Δ*::Ec-MLS-HemH strains. Consistent with protoporphyrin IX being the pink pigment in the Sc-*hem15Δ*::Ec-HemH strain, the Sc-*hem15Δ*::Ec-HemH *hem14Δ* strain lost the pink phenotype, even after growing for 6 days. Moreover, we observed that all strains carrying the *hem14Δ* allele were in fact significantly paler than even wild type BY4741 cells, presumably reflecting extensive protoporphyrin IX depletion in these cells (*Figure 5—figure supplement 2*).

In humans, disrupting heme biosynthesis leads to the disease porphyria, and the secretion of porphyrin intermediates is specifically observed in a subtype known as protoporphyria (*Bloomer et al., 1998*), wherein reduced activity of the human heme pathway protein Hs-FECH leads to accumulation and subsequent secretion of protoporphyrin IX into surrounding tissues. Our data suggest that yeast protein localization and protoporphyrin secretion phenotypes might in the future be exploited to investigate disease-causing mutations in human *Hs-FECH*, even in cases where disease variants do not show any discernible growth defect in yeast.

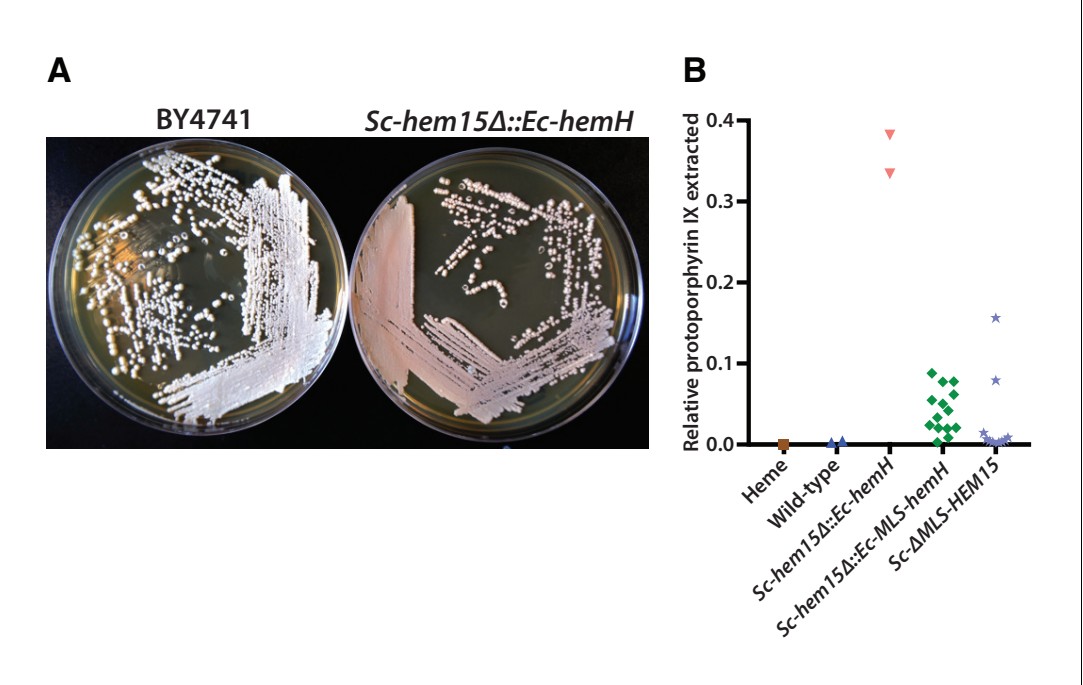

**Figure 5.** Mislocalization of the bacterialized ferrochelatase enzyme identifies a porphyria-like phenotype in yeast. (**A**) Bacterialization of the ultimate yeast gene in the heme biosynthesis pathway results in a distinct pink colony phenotype on YPD agar medium. In contrast, wild type BY4741 strain colonies appear as creamy-white. (**B**) Acetate-extracted secreted products from the pink *Sc-hem15Δ::Ec-hemH* strains show strongly enhanced fluorescence at 635 nm (excitation 399 nm), comparable to a protoporphyrin IX standard and unlike a heme standard or extracts from the parental BY4741 strain. The introduction of an MLS to the bacterialized yeast strain (*Sc-hem15Δ::Ec-MLS-hemH*) significantly reduced protoporphyrin IX secretion, while deletion of the MLS from the native yeast locus in strain *Sc-ΔMLS-HEM15* caused several strains to increase protoporphyrin IX secretion.

The following figure supplements are available for figure 5:

**Figure supplement 1.** Absorbance (top) and emission (bottom) spectra of extracts obtained from acetate (left) and pyridine (right) extraction of the wild type or bacterialized yeast colonies grown on YPD medium.

**Figure supplement 2.** Deletion of protoporphyrinogen oxidase, Sc-HEM14, in the Sc-*hem15Δ::Ec-hemH* strain suppressed the porphyria-like pink phenotype.

## Most yeast heme biosynthesis enzymes can also be successfully plantized

The data above show that genes in the yeast heme biosynthesis pathway can be replaced by their bacterial counterparts, extending earlier studies demonstrating that some heme biosynthesis genes can also be humanized (*Kachroo et al., 2015*; *Sun et al., 2016*; *Schauer and Mattoon, 1990*). Given the ancient conservation of this pathway, we sought to further expand our investigation of its replaceability by swapping the corresponding genes from the plant *Arabidopsis thaliana* into yeast. In plants, heme biosynthesis enzymes form precursors for chlorophyll, and the pathway is largely chloroplast-localized, in contrast to compartmentalization of the heme biosynthetic pathway between the mitochondria and cytosol in many other eukaryotes (*UniProt Consortium, 2015*; *Ashburner et al., 2000*; *Mochizuki et al., 2010*). Nonetheless, the fact that *Arabidopsis* ferrochelatase was cloned by complementing a mutant yeast phenotype suggests that other heme pathway genes might also successfully replace the yeast genes (*Smith et al., 1994*).

The first enzymatic step in the plant heme biosynthetic pathway is similar to bacteria, a two-step reaction using glutamyl-tRNA as a substrate (*Figure 6A and B*) (*Ilag et al., 1994*). We expressed both plant genes, At-*HEMA1* and At-*GSA2*, simultaneously and were able to functionally replace the

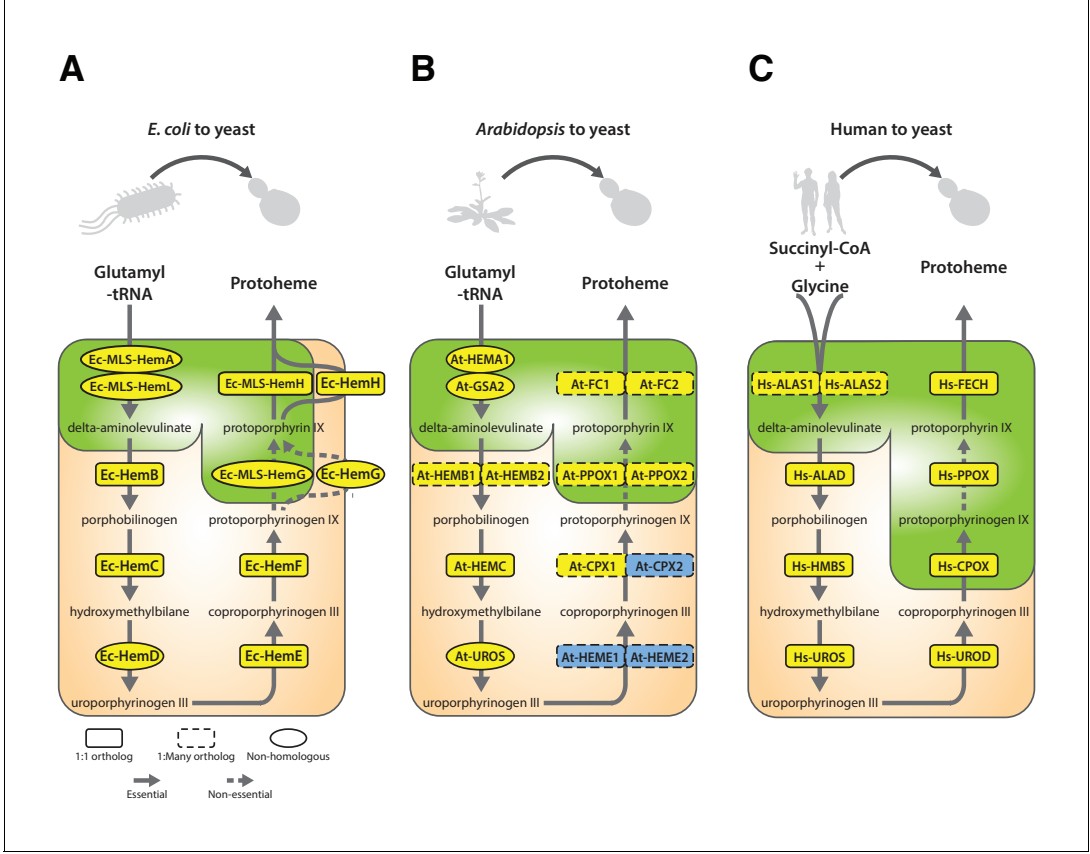

**Figure 6.** Yeast heme biosynthesis pathway enzymes can be successfully replaced by orthologs or analogs from bacteria, plants, and humans, in spite of alterations to subcellular localization. Enzymatic steps of extant bacterial and eukaryotic heme biosynthesis pathways are identical save for the starting metabolites and conversion to delta-aminolevulinate; bacteria also exhibit non-orthologous gene displacement of several enzymes. Heme biosynthesis occurs in the bacterial cytoplasm and inner membrane, the human and yeast in mitochondria and cytoplasm, and the plant in chloroplast and cytoplasm. In spite of these localization changes over evolution, most of the defects in growth rate and viability conferred by heme pathway mutations in yeast can be complemented by introduction of the corresponding (**A**) bacterial genes, (**B**) plant genes (except for At-HemE), and (**C**) human genes. Yellow indicates a replaceable gene, blue non-replaceable.

The following figure supplements are available for figure 6:

**Figure supplement 1.** Heme biosynthesis genes from *Arabidopsis thaliana* and *Glycine max* generally efficiently replace their counterparts in yeast, except in the case of Δ*Sc-Hem12*.

**Figure supplement 2.** Heme biosynthesis enzymes normally localized to plant chloroplasts or human mitochondria localize to the mitochondria when expressed in yeast.

**Figure supplement 3.** Human heme biosynthesis genes efficiently replace their yeast counterparts.

corresponding yeast gene function. Neither protein, when individually expressed, could functionally replace the yeast gene (*Figure 6—figure supplement 1A*).

In *Arabidopsis*, unlike for the case of *E. coli*, a majority of genes in the heme biosynthesis pathway have acquired lineage-specific amplifications, resulting in two co-orthologs for each single yeast gene (*Figure 6B*). In these cases, we tested both co-orthologs individually for replaceability; all replaced successfully, with the exception of one case where only one replaced (*At-CPX1* replaced while *At-CPX2* did not), and one case where neither replaced (*At-HEME1* and *At-HEME2*) (*Figure 6B*, *Figure 6—figure supplement 1B,B'''*).

Because the plant heme biosynthesis pathway builds precursors for chlorophyll synthesis (*Tanaka et al., 2011*; *Papenbrock et al., 1999*), this pathway, especially the penultimate step

producing protoporphyrin IX, is the target of many commercial herbicides. Both *Arabidopsis* paralogs that we tested, *At-PPOX1* and *At-PPOX2*, could efficiently complement the yeast gene responsible for this critical step, *Sc-HEM14* (*Figure 6—figure supplement 1B*). To confirm the generality of these results, we further tested the soybean (*Glycine max*) ortholog *Gm-HEMG* in yeast. As for each of the *Arabidopsis* paralogs, the single soybean ortholog also successfully complemented the *Sc-hem14* deletion growth defect (*Figure 6—figure supplement 1B'*).

It is noteworthy that plant heme biosynthesis genes harbor chloroplast localization sequences (*UniProt Consortium, 2015*), and we did not remove these for our complementation experiments. We speculated that the chloroplast leader peptides might be recognized and localized by the yeast mitochondrial localization machinery, so we constructed EGFP-fusions of the plant enzymes and assayed their localization by fluorescence microscopy. EGFP fusions of At-PPOX1 and At-FC1 showed clear mitochondrial localization in yeast (*Figure 6—figure supplement 2A*). At-FC1 additionally showed amorphous aggregates in some yeast cells, suggesting localization might occasionally be imperfect. Nonetheless, both EGFP-tagged genes were able to efficiently rescue the growth defect of the corresponding yeast gene deletion (*Figure 6—figure supplement 2A*). Thus, these plant chloroplast localization signals appear to be recognized and processed as mitochondrial localization signals in yeast.

These findings suggested that plant versions of cytosolic yeast heme pathway proteins could potentially be mis-localizing to the mitochondria in yeast (*Figure 4A*). Indeed, *At-HEMC* only weakly replaced the yeast gene, *Sc-HEM3*. We found that removing the chloroplast localization signal (CLS) from *At-HEMC* markedly enhanced its ability to functionally replace its yeast ortholog (*Figure 6—figure supplement 1B''*). In contrast, neither of two *Arabidopsis* paralogs, *At-HEME1* and *At-HEME2*, could functionally replace their yeast ortholog, *Sc-HEM12*, even after removing their CLS sequences, or even when co-expressed in the yeast strain (*Figure 6—figure supplement 1B'''*). We speculate that there could be several other reasons why complementation failed, including unknown intermediate reactions, required localization in a special compartment (e.g. chloroplast) or different transcriptional/translational regulation in plants that might contribute to the lack of functional replaceability.

## Each yeast heme biosynthesis enzyme can be replaced by its human ortholog

Earlier studies have shown successful replacement of the yeast heme biosynthesis genes by their human orthologs *Hs-ALAD* (*Schauer and Mattoon, 1990*), *Hs-HMBS, Hs-CPOX and Hs-FECH* (*Kachroo et al., 2015*), while *Hs-UROS* expression resulted in toxicity and *Hs-UROD* failed to replace its yeast ortholog (*Kachroo et al., 2015*; *Sun et al., 2016*). We, therefore, sought to complete tests of the remaining human genes in the pathway. In the case of *Hs-UROS*, we reasoned that toxicity was due to expression from the heterologous constitutive promoter (*Figure 6—figure supplement 3A*). Indeed, similar to the results obtained with the yeast version of this gene (*Figure 4—figure supplement 1*, *Sc-HEM4*), we found that toxicity could be abrogated by inserting the human gene at the native yeast chromosomal locus, thus providing native yeast gene expression and regulation for the human ORF (*Figure 6—figure supplement 3B*). This suggests that, at least in yeast, this step is regulated transcriptionally for optimal function. We also found that the human ORFeome clone of *Hs-UROD* contained a mutation (G303V) that when reverted to wild-type sequence allowed it to replace the yeast gene (*Figure 6—figure supplement 3C and D*), and we additionally confirmed that human *Hs-PPOX* could complement the severe growth defect of the yeast *Sc-hem14* deletion strain (*Figure 6C*, *Figure 6—figure supplement 3D*). Finally, in humans, the initial step of heme biosynthesis is identical to that of yeast (*Sc-HEM1*) but is encoded by two co-orthologs, *Hs-ALAS1* and *Hs-ALAS2*. We found that both of these human genes could individually replace the yeast gene function (*Figure 6C*, *Figure 6—figure supplement 3D*).

The subcellular localization of heme biosynthesis differs slightly between humans and yeast, such that the last three proteins in the human heme biosynthesis pathway are mitochondrially localized, as opposed to only the last two in yeast (*Grandchamp et al., 1978*; *Ferreira et al., 1988*). As all three of these genes replaced, we tested if the human genes were localized to the mitochondria in yeast. Indeed, EGFP-tagged Hs-FECH, Hs-PPOX, and Hs-CPOX all localized to mitochondria in yeast (*Figure 6—figure supplement 2B*) and efficiently rescued the growth defect of the corresponding yeast gene deletion (*Figure 6—figure supplement 2B*), confirming that the human mitochondrial

localization signal is recognizable by the yeast localization machinery. Thus, across our attempts to humanize, plantize, and bacterialize this pathway, the presence of mitochondrial leader peptides from the human genes and the chloroplast leader peptides from the plant genes, as well as the absence of bacterial leaders, all overrode the native yeast localization of the heme biosynthesis pathway. However, the pathway function was largely resilient to these effects, with the exception of protoporphyrin IX accumulation in the mislocalized bacterialized strains (*Figure 5*).

## Heme biosynthesis is a near-universally swappable pathway

As illustrated in *Figure 7*, the heme pathway has had a complicated evolutionary trajectory in eukaryotes due to endosymbiotic events, which has served to increase its similarity between bacteria and eukaryotes (*Kořený and Oborník, 2011*). During eukaryogenesis, early eukaryotes adopted a large portion of the bacteria-like heme biosynthesis pathway of their endosymbiont mitochondria. The subsequent endosymbiotic acquisition of chloroplasts along the plant lineage (*Oborník and Green, 2005*) resulted in redundancy between mitochondrial-origin and chloroplast-origin portions of their heme biosynthesis pathways, a state that can be observed today in *Euglena*, a non-plant,

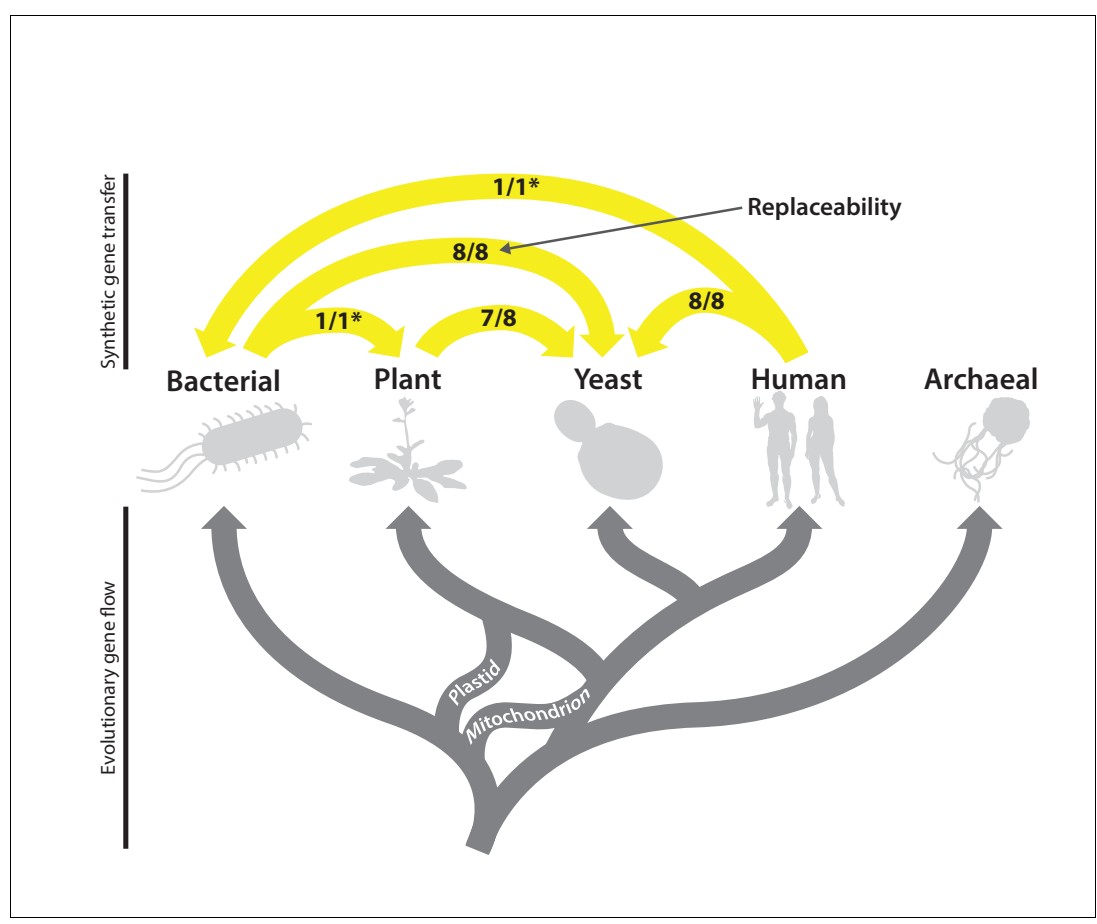

**Figure 7.** The complex evolutionary history of the heme biosynthesis pathway is reflected in high replaceability across species. In eukaryotes, heme biosynthesis enzymes have been replaced historically by endosymbiosis events from bacteria, leading to higher similarity across these lineages, while the archaeal pathway appears to be more divergent (*Storbeck et al., 2010*). Following the endosymbiosis of the cyanobacterial chloroplast, plants adopted most of the chloroplast-derived heme biosynthesis genes, losing many ancestral eukaryotic heme pathway genes (*Oborník and Green, 2005*). Yeast and humans both retain the predicted ancestral eukaryotic heme biosynthesis pathway. While enzymatic steps are mostly shared between yeast, plants, bacteria, and humans, localization of individual proteins differs substantially between species. Asterisks indicate results curated from literature.

photosynthetic eukaryote with more recently acquired chloroplasts (*Kořený and Oborník, 2011*). Over time, plants kept the chloroplastic system and lost most of the mitochondrial system. These evolutionary transfers may have been possible due the apparent modularity of the heme pathway, which we observe in its high tolerance for substituting genes or enzymatic functions across species.

Our data demonstrate that despite 2 billion years of divergence from their last common ancestor, heme biosynthesis genes are still carrying out a conserved and necessary function that can be swapped into yeast with minimal effect on growth and irrespective of orthology and subcellular localization. Taking these data together with literature studies showing successful replacement of the *E. coli Ec-hemG* gene by the plant or human Hs-PPOX gene (*Lermontova et al., 1997*; *Dailey and Dailey, 1996*; *Narita et al., 1996*), and that introducing the protoporphyrinogen oxidase from *Bacillus subtilis* into plants improves yields (*Lee et al., 2000*), heme biosynthesis thus appears to be a pathway whose genes are freely exchangeable between bacteria, plants (with the exception of At-HEME), humans, and yeast (*Figure 7*).

## Conclusions

In conclusion, in order to discern whether orthology strictly confers function across deep evolutionary distances, we systematically tested *E. coli* genes with 1:1 orthology to essential yeast genes for their ability to functionally replace their yeast counterparts. We discovered that ~61% (31/51) of the tested *E. coli* and yeast genes still retain ancestral function to a sufficient extent that the bacterial genes efficiently replace their yeast equivalents. Eukaryote-specific features such as subcellular localization (4 of 14) and proper start codon usage (2 of 4) were critical for swappability for some of the *E. coli* orthologs. Our analysis of replaceable/non-replaceable orthologous pairs revealed that amino acid sequence similarity was not the most important property, consistent with a general trend for sequence conservation to often more strongly reflect other attributes of protein function (e.g., abundance and protein-specific functional constraints) (*Jordan et al., 2002*; *Wang and Zhang, 2009*). Rather, the top predictors of replaceability were features attributed to specific gene modules. These results largely agree with previously published work on humanization of yeast genes (*Kachroo et al., 2015*; *Hamza et al., 2015*; *Sun et al., 2016*), suggesting that functional replaceability is predominantly determined at the level of pathways and processes, even across very large evolutionary distances. As our assays can be considered a form of forced horizontal gene transfer, our results provide support for the 'complexity hypothesis' (*Jain et al., 1999*), which posits that informational (transcription, translation, etc.) genes are less likely to be horizontally transferred than those genes that are operational (metabolism, housekeeping, etc.). Consistent with this expectation, we see metabolism-associated genes replacing more often than those involved in 'informational' processes like transcription or translation.

In the course of these studies, we found that heme biosynthetic reactions were entirely replaceable across the prokaryote-eukaryote divide, despite non-orthologous functional displacement and lack of eukaryotic subcellular localization by native *E. coli* genes (*Figure 7*). Although the archaeal pathway is considerably diverged, our studies across bacteria and eukaryotes showed a high degree of replaceability: Plant heme biosynthesis enzymes functionally replaced yeast enzymes in all but one reaction. Swaps of the corresponding human enzymes into yeast in this and prior studies all suggest that heme biosynthesis is a near universally replaceable pathway.

Our results thus demonstrate that orthologous genes carry out similar functions that allow for their ability to functionally replace each other across even the 2 billion year evolutionary rift separating prokaryotes and eukaryotes from their last common ancestor. These swaps allow engineering of orthologous pathways in model organisms highly amenable to genetic perturbations, like yeast and bacteria, for further characterization.

## Materials and methods

### Construction of ORFs from bacteria, plants, and humans in yeast expression vectors

Refer to *Supplementary file 3* for all the primers used in this study.

### E.coli ORF yeast expression vectors

Initial *E. coli* ORF primers were designed such that the 3' ends of the primers had homology to *E. coli* genes and 5' ends contained a universal flanking sequence. A second round of PCR was performed with primers recognizing the universal flanking sequence and also having 5' ends corresponding to gateway compatible attL1 (or attB1) and attL2 (or attB2) sequences on the forward and reverse primers, respectively. Resulting PCR products from attL sequence containing primers were directly cloned via gateway LR cloning (ThermoFisher Scientific) into yeast destination vector pAG416GPD-ccdB (Addgene) to create expression clones. PCR products from attB primers were subcloned via gateway BP cloning into vector pDONR221 (ThermoFisher Scientific) to create entry clones. These entry clones were then cloned via gateway LR to the pAG416GPD-ccdB destination vector to create expression clones. Some *E. coli* genes were synthesized as gBlocks from IDT and made gateway compatible by adding attL1 and attL2 sequences at the 5' and 3' ends, respectively, making them compatible for direct LR cloning to create expression vectors.

### Plant ORF yeast expression vectors

*Arabidopsis thaliana* ORFs were PCR amplified from cDNA obtained as a kind gift from Dr. Jeffrey Chen (UT Austin), using primers specific to each gene and containing gateway compatible attL1 and attL2 sequences at the 5' and 3' ends respectively (*Supplementary file 3*). PCR products were directly cloned into the yeast expression vector pAG416GPD-ccdB by LR gateway cloning (using LR clonase II from Invitrogen). *At-HEME1* and *At-HEMB2* were synthesized as gBlocks from Integrated DNA Technologies (IDT).

### Plant ORF yeast expression vectors without the chloroplast localization signal

In order to remove the chloroplast localization signal from the plant proteins At-HEMC, At-HEME1 and At-HEME2, we first performed amino acid sequence alignment with the bacterial and yeast orthologs to identify unaligned N-terminal sequence. We attributed the non-alignment to the presence of chloroplast localization signal (CLS) sequence. We also used the TargetP 1.1 signal peptide predictor (*Emanuelsson et al., 2007*) to corroborate the sequence alignments. In the case of At-HEMC, 68 N-terminal amino acids were deleted while retaining ATG start codon. Similarly, in the case of At-HEME1 and At-HEME2, 47 N-terminal amino acids were deleted while retaining the ATG start codon. We synthesized these genes as gBlocks (IDT) with attB1 and attB2 attachment sites flanking their 5' and 3' ends, respectively, then subcloned the gBlocks into the entry clone pDONR221, sequence-verified the clones, and LR cloned the genes into yeast expression vector pAG416GPD-ccdB.

### Plant ORF yeast expression vectors for co-expression of At-HEME1 and At-HEME2

*At-HEME1* and *At-HEME2* were cloned (with or without CLS) into the destination vectors pAG416GPD-ccdB and pCMY41 (kind gift of Christopher Yellman; pCMY41 is identical to pAG416GPD-ccdB but carries a hygromycin-resistance cassette), allowing us to co-transform two plasmids and select for the double plasmid transformants on synthetic defined medium, -Ura + Hyg (200 μg/ml).

### Human ORF yeast expression vectors

Human ORF's were obtained from the ORFeome collection (GE Dharmacon) and sequenced to verify correct, full-length clones. In the case of human *Hs-UROD*, the ORFeome clone contained a loss-of-function mutation (G303V), so wild-type human *Hs-UROD* was synthesized as a gBlock fragment (IDT) and used as a PCR template, amplifying the gene using primers with flanking gateway compatible sites attL1 and attL2 at the 5' and 3' ends respectively (*Supplementary file 3*). The PCR product was subcloned by LR reaction into the yeast expression vector pAG416GPD-ccdB.

### Yeast ORF yeast expression vectors

Yeast ORFs were amplified using PCR from genomic DNA of yeast strain BY4741, and gateway compatible attL1 and attL2 sequences added by PCR to the amplicons 5' and 3' ends, respectively (*Supplementary file 3*). The resulting PCR products were subcloned by LR reaction into the yeast expression vector pAG416GPD-ccdB. Several yeast heme biosynthesis genes were first cloned in pENTR/SD/D-TOPO plasmid (Invitrogen) to obtain gateway entry clones (refer to *Supplementary file 3* for primers). These clones were sequence-verified and then used to generate yeast expression vectors by LR reaction into the vector pAG416GPD-ccdB.

### Mitochondrially-localized *E. coli* ORF yeast expression vectors

We added the MLS from the yeast MIP1 gene to the 5' end of selected *E. coli* ORFs via PCR, using an ORF-specific ultramer containing the full MLS-coding sequence at the 5' end such that MLS was in frame with the coding sequence of the *E. coli* gene while removing the *E. coli* gene start codon (*Supplementary file 3*). Each PCR product was then used as a template to add gateway cloning attachment sites attL1 and attL2, followed by LR gateway cloning into pAG416GPD-ccdB to generate yeast expression vectors.

### EGFP tagged *E. coli* / plant / human ORF yeast expression vectors

Using PCR, we amplified *E. coli* / plant / human ORFs without their respective stop-codons while also adding attB1 and attB2 gateway attachment sites at the 5' and 3' ends of each PCR product (*Supplementary file 3*). The resulting PCR fragments were subcloned into plasmid pDONR221 to generate gateway entry clones using the BP gateway cloning reaction. Each entry clone was subjected to the LR cloning reaction in order to generate a carboxy-terminal EGFP-tagged yeast expression clones in the pAG416GPD-ccdB-EGFP destination vector.

### Converting *E. coli* ORF yeast expression vectors with alternative start codons to ATG start codon

We introduced ATG start codons by PCR mutagenesis, employing ATG-containing primers (*Supplementary file 3*) to amplify and simultaneously add gateway cloning attachment sites attL1 and attL2 to the 5' and 3' ends of the PCR products, respectively, then subcloning these products by the LR gateway cloning reaction into the pAG416GPD-ccdB plasmid in order to construct yeast expression vectors.

### *E.coli* and Arabidopsis two-gene expression vectors for complementing a yeast Sc-HEM1 deletion

*E. coli* genes *Ec-hemA*, *Ec-hemL* and plant genes *At-HEMA1*, *At-GSA2* were PCR amplified from genomic DNA (*E. coli*) or gBlocks obtained from IDT (*Arabidopsis*). For *E. coli* genes, we also added an MLS at the 5' end of the PCR products. These PCRs were made Golden Gate compatible by introducing Bsmb1 sites and cloned individually in pYTK001 (*Supplementary file 3*). In the case of *At-HEMA1*, the gBlock was synthesized to mutate an internal BsmBI site such that it doesn't affect the protein sequence. Clones were sequence verified prior to assembly (*Lee et al., 2015*). Individual transcription units for each of the genes were obtained by Golden Gate assembly using the pYTK001-entry clone containing the *E. coli* or plant gene, along with pYTK vectorscontributing promoters and terminators. In the case of *Ec-hemA* and *At-HEMA1* transcription units (TU1's), the pHHF2 promoter was contributed by pYTK012 and tADH1 terminator by pYTK053. In the case of *Ec-hemL* and *At-GSA2* transcription units (TU2's), the pTEF1 promoter was contributed by pYTK013 and tSSA1 terminator by pYTK052. Unique contigs for directional assembly were obtained from pYTK002 (ConLS) and pYTK067 (ConR1) for TU1. For TU2, the unique contigs were obtained from pYTK003 (ConL1) and pYTK072 (ConRE). The individual transcription units (TU1 and TU2) were then assembled in a single yeast CEN6-URA vector via Golden Gate assembly with BsmbI.

All clones were sequence-verified using the University of Texas Genomic Sequencing and Analysis Facility.

## Functional complementation assays

Gene replaceability was tested using available yeast strains from two yeast strain collections, the temperature-sensitive (TS) collection (*Li et al., 2011*) and the heterozygous diploid deletion magic marker collection (*Pan et al., 2004*), as follows:

### (1)Temperature-sensitive (TS) collection assays

Typically, yeast strains in this collection grow at permissive temperatures (22–26°C) but cannot grow at restrictive temperatures (35–37°C). Growth at restrictive temperatures thus allows for the identification of foreign genes that complement the yeast defect. We tested for replaceability in temperature-sensitive yeast strains as follows:

The strains were transformed with either an empty vector control (pAG416GPD-ccdB) or with the clone expressing the foreign gene. The transformants were plated on:

1. Ura dextrose medium at the permissive temperature (25°C), serving as a control for transformation efficiency and/or toxicity since both the yeast and the human gene are expressed.
2. Ura dextrose medium at the non-permissive temperature (36°C), testing for functional replacement under conditions in which the corresponding yeast gene is non-functional.

### (2)Heterozygous diploid deletion magic marker collection assays

The yeast heterozygous diploid deletion magic marker collection comprises yeast strains that harbor a deletion of one copy of a yeast gene replaced with a KanMX cassette. The strains also carry a magic marker or synthetic genetic array (SGA) cassette at the *can1* locus, which enables selection for haploid cells on magic marker (MM) medium (−His −Arg −Leu +Can) post-sporulation with or without antibiotic G418 (200 µg/ml). Haploid a-type spores that harbor a wild type gene grow normally on magic marker (MM) medium without G418 and provide a test of sporulation efficiency and toxicity, if any, associated with heterologous expression of the foreign gene (using a −Ura selection marker in this study). Growth of haploid spores on MM medium in the presence of G418 selects for yeast cells that harbor the relevant gene deletion while testing for complementation by the foreign gene.

Expression clones or empty vector controls were transformed into appropriate strains and selected on −Ura G418 medium in a 96-well format. (Toxicity was inferred from a repeated failure to obtain transformants in the case of expression clones compared to the empty vector control) Transformants were re-plated on GNA-rich pre-sporulation medium containing G418 (200 µg/ml) and histidine (50 mg/l). Individual colonies were inoculated in liquid sporulation medium containing 0.1% potassium acetate, 0.005% Zinc acetate, and incubated with vigorous shaking at 25°C for 3–5 days, after which sporulation efficiency was estimated by microscopy, and the mixture re-suspended in water and equally plated on two assay conditions:

1. 'G418 minus' magic marker dextrose medium (−His −Arg −Leu +Can −Ura), incubated at 30°C. The haploid spores that carry the wild-type yeast gene grow in this medium acting as a control for sporulation efficiency. This condition also assays for toxicity if the haploid spores carrying expression vectors fail to grow.
2. 'G418 plus' magic marker dextrose medium (−His −Arg −Leu +Can −Ura) containing 200 µg/ml G418. The resulting haploid deletion strain is expected not to grow, providing an assay of replaceability for strains carrying the expression vector. Cases with approximately equal numbers of cells growing in the absence or presence of G418 were considered functional replacements.

Positive assays were verified independently. Individual colonies were isolated from selective plates and used for growth assays on YPD or magic marker medium with G418 (*Figure 1B*, *Figure 1—figure supplement 1A*). After growth on YPD with G418, each strain was spotted on 5-FOA agar to test plasmid dependency (*Supplementary file 1*). Only one strain (*Ec-valS*) failed that test.

## Ortholog inference

Genes with 1:1 orthology between yeast and *E. coli* were obtained from the Inparanoid 8 webserver (*Sonnhammer and Östlund, 2015*) and filtered to an only yeast-essential set. Orthologs to these selected yeast genes in human and *Arabidopsis* were downloaded from Inparanoid 8 and further

refined by comparison to orthology calculations by eggNOG4.5 (*Huerta-Cepas et al., 2016*), OMA (*Altenhoff et al., 2015*), and reference to the evolutionary history of the heme pathway in photosynthetic organisms (*Oborník and Green, 2005*).

## Computational analyses of replaceability

### Feature assembly

### Sequence features

Protein sequence features were calculated using UniProt (*UniProt Consortium, 2015*) proteomes from the respective species downloaded in March 2015. *E. coli* nucleotide sequence features were calculated using EcoGene (*Zhou and Rudd, 2013*) sequences downloaded April 2015.

[Sc|Ec]_Length

The number of amino acids in the respective protein.

Sc-Ec_LengthDifference

Calculated as the difference of the amino acid length of the *E. coli* protein subtracted from the length of the *S. cerevisiae* ortholog.

Sc-Ec_AbsLengthDifference

Calculated as the absolute value of the above length difference.

Sc-Ec_PercentIDAligned

Sc-Ec_PercentIDLongest

Sc-Ec_PercentSimilarityAligned

Sc-Ec_PercentSimilarityLongest

The fraction of identical residues (PercentID) or similar residues (PercentSimilarity) in a global alignment (NWalign, http://zhanglab.ccmb.med.umich.edu/NW-align/) of the respective orthologs, as a function of the longest of the two (Longest) or the length of the aligned region (Aligned).

Ec_CAI

Ec_CBI

Ec_FOP

Ec_ScCAI

Ec_ScCBI

Ec_ScFOP

The Codon Adaptation Index (CAI), Codon Bias Index (CBI), or Frequency of OPtimal codons (FOP) for the respective *E. coli* gene, calculated using the E. coli optimal codon table (Ec_) or *S. cerevisiae* optimal codon table (Ec_Sc) using codonw (http://sourceforge.net/projects/codonw/).

### Abundance features

Sc_TranscriptAbundance

Sc_ProteinAbundance

Sc_RPFAbundance

Sc_TranslationEfficiency

Ec_ProteinAbundance

Yeast (Sc) protein abundance data was taken from *Kulak et al. (2014)*. Yeast Transcript and RPF abundance were taken from *Ingolia et al. (2009)*. Yeast RPF abundance is calculated as the ratio of RPF reads to Transcript reads for a given gene. *E. coli* data was taken from *Arike et al. (2012)* (average iBAQ abundance only).

### Network features

Sc_BIOGRID-Betweenness

Sc_BIOGRID-Clustering

Sc_BIOGRID-Degree

Sc_BIOGRID-SumLLS

Sc_BIOGRID-LT-Degree

Sc_BIOGRID-LT-SumLLS

Sc_BIOGRID-LT-Betweenness

Sc_BIOGRID-LT-Clustering

Calculated from interactions present in BIOGRID 3.1.93 (*Stark et al., 2006*). 'BIOGRID' was calculated using only those interactions annotated as 'physical interactions', while 'BIOGRID-LT' was calculated using the subset of physical interactions found only by low-throughput experiments.

Ec_EcoCyc_FractionComplementing

Calculated using the 'All Pathways' table from EcoCyc (https://ecocyc.org) downloaded in September 2016. To create the network, all pathways were considered 'cliques' so that all members of the pathway were annotated as interacting with all other members of the pathway. FractionComplementing is the fraction of interacting partners tested in our assays that were able to replace.

## Calculating the predictive strength of features

The predictive power of each feature was calculated as the area under the receiver-operator characteristic curve (AUC) while treating each feature as an individual classifier. Each feature was sorted in both ascending and descending directions, retaining the direction providing an AUC > 0.5. To assess significance, a shuffling procedure was performed as follows: For each feature, the replaceable/non-replaceable status of each ortholog pair was shuffled (retaining the original ratio of replaceable to non-replaceable assignments), and the AUC was calculated. The shuffling procedure was carried out 1000 times for each feature, and the mean AUC values and their standard deviations are reported.

## Combined classifier

A Random Forest classifier was constructed using all features and evaluated using 10-fold cross-validation. The random forest was constructed to have no maximum tree depth, and ties between similarly good attributes were broken randomly. The combined classifier was implemented using the Weka data-mining software (*Frank et al., 2004*).

## Confocal microscopy

Yeast cultures expressing GFP-tagged bacterial, plant, or human genes were grown to an optical density (OD) of ~1, then 500 µl of the culture washed with 1X PBS, and mitochondria fluorescently labeled by adding 100 nM MitoTracker Red CMXRos (Invitrogen). The cells were incubated in the dark on a mildly shaking platform for 20 min at room temperature, then washed twice with 1X PBS and resuspended in 15 µl of 1X PBS for imaging by confocal microscopy, using a Zeiss LSM 710 confocal microscope with a Plan-Apochromat 63×/1.4 oil-immersion objective.

## Quantitative growth curves

Yeast strains were either pre-cultured in liquid YPD or -Ura Dextrose selective medium for 2 hr or overnight respectively. The culture was diluted in YPD or -Ura Dextrose medium to an OD of ~0.1 in 100 or 150 µl total volume in a 96-well plate. Plates were incubated in a Synergy H1 shaking incubating spectrophotometer (BioTek), measuring the optical density every 15 min over 48 hr. Growth curves were performed in triplicate for each strain by splitting the pre-culture into three independent cultures for each 48–60 hr time course.

## Detection of heme pathway intermediate metabolites

Bacterialized *Ec-hemH* yeast strains were grown on YPD as lawns or large patches for 5 days (the phenotype manifests after several days of growth). Clumps of cells about 5–7 mm in diameter were collected with a toothpick and first suspended in water, then pelleted at 15,000 g for 30 s. This created a distinctive pale yellow yeast pellet, with the red pigment appearing in a small clump on top. The water was removed while carefully avoiding disruption of the red pigment pellet, after which we performed extractions with two different methods. The first method, based on Bassel et al. (*Bassel et al., 1975*), was to add 1 ml pyridine to each pellet, spinning down at 15,000 g for 30 s and recovering only the liquid fraction (cell debris would pellet down while the red pigment migrated into the liquid pyridine phase). The second referred to as 'acetate extraction' in this text, was to extract with a 3:1 ethyl acetate:glacial acetic acid solution as described in *Pretlow and Sherman (1967)*.

We then measured the absorbance of the extractions in a transparent plastic 96-well plate on the (Synergy H1 from BioTek) on wavelengths from 223 nm to 998 nm, with 1 nm steps. We measured

fluorescence on the same instrument by exciting at 399 nm and measuring emission at 450 nm to 699 nm with 1 nm step. The spectra were compared with those shown in *Bark et al. (2010)*.

We also obtained protoporphyrin IX (*Sigma-Aldrich*, P8293-1G) and hemin B (*Sigma-Aldrich*, 51280–1G) and suspended these in acetate and pyridine to closely resemble the chemistry of our extractions. These solutions were measured alongside the extractions themselves as standards, in order to further confirm the identity of the molecules we detected.

## Replacement of bacterial and human genes at their native yeast loci using CRISPR-Cas9

Genomic editing and replacement of yeast ORFs is described in greater detail at Bio-protocol (*Akhmetov et al., 2018*).

### Bacterializing yeast strains at native genomic loci using CRISPR

We inserted *E. coli* ORFs at their native yeast loci using CRISPR/Cas9-mediated double strand breaks (DSB) and homologous recombination. The integration was performed by chemically co-transforming yeast with a linear template DNA (Zymo Research - #T2001) and a plasmid carrying Cas9 and gRNA targeting the desired locus of integration (refer to *Supplementary file 3*). The transformed cells were plated on SD-Ura medium to select for successful transformation of the plasmid (CRISPR-induced DSBs act as partial selection against background), and screened for successful integration of the template via colony PCR using primers flanking the start codon of the ORF (a forward primer annealing to the promoter and a reverse primer annealing to the *E. coli* ORF) (*Figure 4—figure supplement 4*).

The template DNA is a linear sequence containing the *E. coli* ORF, flanked by the yeast promoter and terminator which act as homology. In order to produce this template DNA, we designed primers for each gene that amplify the entire coding sequence of the *E. coli* ortholog, while also inserting flanking homologies to the yeast locus targeted. In most cases, we used primers 120 bp long, with about 20 bp shared with the *E. coli* gene and 100 bp of yeast homology. In cases where this template failed to integrate (such as *Ec-hemC*) we designed 200 bp primers with about 180 bp homology. For chimeric ORFs of *E. coli* genes *Ec-hemG* and *Ec-hemH* that retained the native yeast MLS, the template was produced by including the MLS in the forward primer sequence. We amplified the template DNA with PCR, purified it using the DNA Clean and Concentrator-25 kit (Zymo Research - #D4006); final elutions were done with water. We used 5 µg DNA template per transformation, in cases where this failed we attempted it again with 10 µg.

CRISPR plasmids were constructed using a Golden Gate-based cloning strategy as described in *Lee et al. (2015)*. Briefly, for each yeast gene we designed two gRNA sequences using Geneious v9 (*Kearse et al., 2012*); both sequences were selected from within the yeast ORF so as to exhibit high predicted efficiency with a low background activity for the rest of the yeast genome. We performed integration experiments separately for each gRNA, as often one of the gRNA sequences would have substantially lower efficiency than predicted. As per Lee et al. (*Lee et al., 2015*), each gRNA sequence was first synthesized as an oligonucleotide (IDT), subcloned into intermediate plasmids, and eventually into a Cas9 plasmid carrying a Ura selectable marker, finally transforming 500 ng into yeast cells for the integration assay.

In order to construct yeast strain *Sc-ΔMLS-HEM15*, we started with *Sc-hem15Δ::Ec-hemH* yeast which had lost their CRISPR plasmid, and co-transformed them with CRISPR plasmids carrying gRNA that targets the *Ec-hemH* sequence, as well as template DNA created by amplifying the yeast *Sc-HEM15* sequence from yeast genomic DNA. The MLS was deleted by designing template amplification primers which leave it out. This was necessary since the MLS sequence did not contain unique CRISPR targets, thus it was not possible to construct a CRISPR system that would cleave wild type *Sc-HEM15* but not the desired *Sc-ΔMLS-HEM15*.

### Humanizing Hs-UROS gene at the native yeast locus

We co-transformed the plasmid expressing Cas9 and gRNA targeting yeast *Sc-HEM4* gene and repair PCR template that contains human *Hs-UROS* gene flanked by 100 bp of homologous sequence to the yeast *Sc-HEM4* promoter and terminator region. The colonies that grew after the transformation of CRISPR plasmid and the repair template were verified for the human gene

insertion using a forward primer outside the region of homology and reverse primer specific to the human gene. The positive PCR reaction with appropriate size (375 bp) confirmed the right clone.

## Generation of Sc-*HEM14* yeast deletion strains

Using CRISPR, we deleted the *Sc-HEM14* ORF in wild type BY4741, Sc-*hem15Δ*::Ec-*HemH,* and Sc-*hem15Δ*::Ec-MLS-*HemH* strains. Specifically, we co-transformed the plasmid expressing Cas9 and gRNA targeting the yeast *Sc-HEM14* gene with a 200 bp oligonucleotide repair template comprising 100 bp each of sequence matching the 5' and 3' UTRs of the *Sc-HEM14* gene and selected for growth on SD-Ura medium. The resulting *hem14Δ* strains were confirmed by PCR using primers outside the region of homology. *Supplementary file 3* provides relevant primers and oligos.

## Acknowledgements

This work was supported by grants from the NIH (R21 GM119021, R01 HD085901, DP1 GM106408, R01 DK110520, R35 GM122480), CPRIT, and the Welch foundation (F-1515) to EMM.

## Additional information

### Funding

| Funder | Grant reference number | Author |
|---|---|---|
| National Institutes of Health | R21 GM119021 | Edward M Marcotte |
| Cancer Prevention and Research Institute of Texas | | Edward M Marcotte |
| Welch Foundation | F1515 | Edward M Marcotte |
| National Institutes of Health | R01 HD085901 | Edward M Marcotte |
| National Institutes of Health | DP1 GM106408 | Edward M Marcotte |
| National Institutes of Health | R01 DK110520 | Edward M Marcotte |
| National Institutes of Health | R35 GM122480 | Edward M Marcotte |

The funders had no role in study design, data collection and interpretation, or the decision to submit the work for publication.

### Author contributions

AHK, JML, Conceptualization, Data curation, Formal analysis, Supervision, Validation, Investigation, Visualization, Methodology, Writing—original draft, Writing—review and editing; AA, Data curation, Validation, Investigation, Visualization; MS-J, Validation, Investigation, Visualization; CDM, Visualization, Methodology, Writing—review and editing; AZ, Visualization, Methodology; EMM, Conceptualization, Resources, Data curation, Supervision, Funding acquisition, Investigation, Visualization, Methodology, Writing—original draft, Project administration, Writing—review and editing

### Author ORCIDs

Aashiq H Kachroo, http://orcid.org/0000-0001-9770-778X
Jon M Laurent, http://orcid.org/0000-0001-6583-4741
Claire D McWhite, http://orcid.org/0000-0001-7346-3047
Edward M Marcotte, http://orcid.org/0000-0001-8808-180X

## Additional files

### Supplementary files

• Supplementary file 1. Detailed results of complementation assays. Rows are ortholog pairs. Columns A-E list several alternative gene IDs for each organism. 'Assay location' is the location of that complementation assay in the plate images shown in *Figure 1—figure supplement 1*. The following columns list the results of the specific assay type: 'MM' refers to the heterozygous diploid assay (Magic Marker). 'TS' refers to the temperature-sensitive allele assay. 'With MLS' are assays re-done

with the yeast mitochondrial localization sequence as described in the test. 'With ATG' are assays redone with an ATG start codon substituted for the *E. coli* genes non-canonical start codon. 'Plasmid dependence' lists the results of 5' FOA screening of complementing clones to confirm that the *E. coli* gene-containing plasmid is present. 'Final status' and 'Preliminary status' refer to the complementation status of the given ortholog pair after (Final) or before (Preliminary) accounting for MLS, ATG, or plasmid dependence assays.

• Supplementary file 2. Data used to calculate predictive features. The first sheet displays all data for each property used in the study to determine predictive properties. The first several columns list IDs for the two organisms as well as the final complementation results. The following columns include all data for each feature. See Materials and Methods for detailed descriptions of each property. The second sheet lists the calculated AUCs for each feature, and the results of the shuffling procedure for each.

• Supplementary file 3. Primers used in this study. A brief description of each primer's use is included on each separate sheet of the file. For additional information, see Materials and methods for the relevant section.

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
