## [Decision Letter]

Thank you for submitting your article "Systematic bacterialization of yeast genes identifies a near-universally swappable pathway" for consideration by *eLife*. Your article has been reviewed by three peer reviewers, and the evaluation has been overseen by Naama Barkai as the Senior Editor and Reviewing Editor. The following individual involved in review of your submission has agreed to reveal their identity: Eugene Koonin (Reviewer #2).

The reviewers have discussed the reviews with one another and the Reviewing Editor has drafted this decision to help you prepare a revised submission.

This paper analyzes systematically the replaceability of yeast gene by their bacterial orthologs and discusses general differences between 'replaceable' and 'non-replaceable' genes. All three reviewers appreciated the study and found it suitable for publication in *eLife*. As you will see, the comments are related mostly to the Discussion and explanations. Please relate and try to address all of these suggestions.

*Reviewer #1:*

In the manuscript titled: "Systemic Bacterialization of yeast genes identifies a near-universal swappable pathway" Kachroo et al. describe their attempt in examining the complementation of 60 essential yeast genes with their bacterial counterparts. Of the 60, for 51 they were able to perform complementation assays. Complementation was tested in a plasmid based expression with the GPD1 promoter, CDS was cloned with the *E. coli* DNA sequence, tested with either the TS library yeast strain or in a heterozygous diploid from the deletion collection. 25 of the 51 complemented their yeast counterparts with no modification. Another 4 of 10 yeast mitochondrial genes complemented once a mitochondrial localization sequence was added. 2 of the 4 genes that lacked the canonical ATG start codon, complemented following mutation of the start codon to ATG. All told 31/51 showed complementation.

The authors then followed this up with statistics about the characteristics that determine replaceability between Bacteria and Yeast, taking into account 22 features, they find that protein sequence similarity is not a highly predictive feature, while specificity of pathway and/or process is highly predictive. Metabolic pathways show high replaceability while highly expressed genes, such as ribosomal genes, are more sensitive to codon choice.

The authors then turn to the heme pathway; they first examine the complementation of each of the bacterial genes for its yeast counterpart. For the first step in yeast (Sc-Hem1) there are two genes in bacteria (Ec-hemA and Ec-hemL) and in yeast it is performed in the mitochondria, they added MLS sequences to both bacterial genes and with co expression showed complementation and localization using a EGFP tag. For two more steps (Sc-HEM4 and Sc-HEM14), there only functional analogs (Ec-hemD and Ec-hemG), not orthologs, and they indeed complemented for the yeast genes. The final two steps in the yeast pathway are carried out in the mitochondria, however, the bacterial genes complemented without an MLS sequence, and surprisingly localized to the plasma membrane. However, when trying to complement for both yeast genes in a single strain there was a fitness defect. When integrating the bacterial genes to express them from their native yeast gene counterparts, all but two complemented, and those two complemented following MLS addition.

In complementation of Sc-HEM15 by Ec-hemH, authors noted that colonies are pink, they analyzed and found this to be due to accumulation of protoprophyrin IX, the substrate of Ec-hemH, determining that mislocalization to the plasma membrane caused reduce enzyme activity and accumulation of the substrate, similarly to the human disease protoporphyria. They suggest that this indicates that the yeast can be used to study mutations in the corresponding human gene.

The authors show that they can also replace most of the yeast heme pathway genes with the *Arabidopsis* genes that encode enzymes that form precursors for chlorophylls; this pathway is localized mostly to the chloroplast. Similarly to the bacterial pathway, the first step from yeast is performed in two steps in *Arabidopsis*, only when both genes (At-HENA1 and At-GSA) were expressed was complementation of Sc-HEM1 observed.

In *Arabidopsis* most of the heme genes are duplicated, in most, individually replaced their yeast orthologs, in one case only one and in one case neither replaced (At-HEME1 and At-HEME2 replacing Sc-HEM12). Due to its interest in commercial herbicides, the step producing protoporphyrin IX was tested for complementation by *Arabidopsis* genes (At-PPOX1 and At-PPOX2) as well as the soybean ortholog, Gm-HEMG; both complemented for Sc-Hem14. The authors also noted that the chloroplast localization signal on a couple of the *Arabidopsis* proteins caused their localization to the mitochondria, and complemented for their yeast orthologs.

Similarly, the authors assayed for complementation by the human orthologs of the heme pathway. Similar to what was shown in previous papers, they saw toxicity when overexpressing Hs-UROS from a plasmid using a constitutive promoter, thus they integrated it to the native locus of Sc-Hem4, and it showed reduced toxicity when integrated. They also showed complementation by Hs-UROD and Hs-PPOX. Two human orthologs complement for Sc-HEM1 individually. Although no yeast MLS was added to the human genes of the last three steps in the pathway they all were found to localize to the mitochondria.

I would consider acceptance of this paper following the authors' response to these issues:

1) The authors do a very poor job of describing the basis for choosing and refining the 60 coli genes. It is really hard to believe that there are only 60 good orthologs between yeast and *E. coli*. What exactly were the criteria for choosing those? Further, they did not indicate what possibly was the issue with the 9/60 that did not have a complementation assay. Was it a problem with the corresponding yeast host strain(s) or was it difficult to clone these genes? This should be spelled out, perhaps as an explicit description in the Materials and methods section. Complete tables of all orthologs attempted for all the donor species tested, along with reasons for failure for the 9 that missed in bacteria for example, should be provided in the supplement.

2) From what I could conclude, in both bacteria and *Arabidopsis* there was an issue with complementing Sc-hem12. The bacteria one could not be integrated and both *Arabidopsis* orthologs failed to complement. What is the issue with Hem12? How come the human ortholog Hs-UROD did complement? Even if there is no single clear answer to these questions a hypothesis would be welcome.

3) For the hemH pink phenotype, to show that this is actually due to the specific substrate proposed, they should delete Sc-HEM14 (or another upstream function) to genetically verify that the pink phenotype disappears because formation of protoprophyrin IX should be prevented by this "upstream" mutation.

*Reviewer #2:*

This is an interesting and important study on the replaceability of yeast gene by 1:1 orthologs from bacteria. To my knowledge, it is the largest systematic analysis of this kind and as such appears to be the best test of the 'Ortholog-Function Conjecture' to be reported so far (see Gabaldon & Koonin 2013, as cited here). The proverbial glass is more than half-full which in itself not particularly surprising but, given the gulf between prokaryotes and eukaryotes, should be considered a resounding vindication of the conjecture.

Apart from the general point above, the main interest of this work lies in the analysis of various predictors and correlates of ortholog replaceability. I do not share the authors' surprise regarding the lack of correlation between replaceability and sequence conservation. It has been shown in a number of analyses that there is at best a very limited connection between sequence conservation and gene essentiality (Jordan IK, Rogozin IB, Wolf YI, Koonin EV. Essential genes are more evolutionarily conserved than are nonessential genes in bacteria. Genome Res. 2002 Jun;12(7):962-8; Wang Z, Zhang J. Why is the correlation between gene importance and gene evolutionary rate so weak? PLoS Genet. 2009 Jan;5(2):e1000329). I think we see here a manifestation of the same phenomenon: sequence conservation depends much stronger on the abundance of a protein product and the gene-specific functional constraints than on the "importance".

The shape of the dependency in Figure 3 seems paradoxical at first glance (non-monotonic curve, with moderately conserved genes being most replaceable) but I suspect is explained by the different in replaceability among functional classes of genes (Figure 3). I find it highly desirable to test this directly and discuss accordingly.

To me, the results in Figure 3 are indeed the most interesting in the paper. Again, this looks striking and at least superficially, paradoxical, in that genes in the most highly conserved categories, such as translation and tRNA modification, are virtually non-replaceable. I believe the explanation lies in the complexity hypothesis (Jain R, Rivera MC, Lake JA. Horizontal gene transfer among genomes: the complexity hypothesis. Proc Natl Acad Sci U S A. 1999 Mar 30;96(8):3801-6) that seems to be the best explanation for the rates of horizontal gene transfer in different functional classes of prokaryotic genes. Indeed, ortholog replacement studied here can be considered an extreme, "forced" variant of horizontal gene transfer. I think a thoughtful discussion of these parallels and their utility for explaining the results could make the present story considerably more interesting.

*Reviewer #3:*

This is a fun and thought-provoking study that systematically replaces essential yeast genes with their orthologs from *E. coli*. There is good rescue by many of the 1:1 orthologs. The authors then more extensively investigate the 'swappability' of all the proteins in the haem biosynthesis pathway, finding that even mislocalised proteins can rescue some yeast deletion phenotypes as can swapping in an alternative (non-orthologous) part of the pathway. This extends their previous work (and the work of two other labs) performing similar experiments with human 1:1 orthologs over a larger evolutionary distance, formally showing that, at least for metabolic enzymes, function has been conserved over huge evolutionary distances to the extent that the enzymes can be swapped from one species to another.

Minor suggestions/queries:

1. The authors do not describe how the two complementation assays work in the main text or how much agreement there is between them when they are directly compared for the same ORFs. I think they should do both to assist the general reader. The rescue of TS mutants is straightforward, but only yeast aficionados will understand the second assay. In addition, this cannot be at all understood from their sentence in the main text: '…were carried out using two types of conditionally essential yeast alleles, consisting of temperature- sensitive (TS) haploid and heterozygous knockout diploid yeast strains”. In Figure 1 and elsewhere it would help to indicate how many ORFs and which were tested with which complementation assay, how many with both and the agreement between them.

2. Are any of the mutant phenotypes not rescued by overexpressing the yeast ORF?

---

## [Author Response]

*Reviewer #1:*

*[…] 1) The authors do a very poor job of describing the basis for choosing and refining the 60 coli genes. It is really hard to believe that there are only 60 good orthologs between yeast and E. coli. What exactly were the criteria for choosing those? Further, they did not indicate what possibly was the issue with the 9/60 that did not have a complementation assay. Was it a problem with the corresponding yeast host strain(s) or was it difficult to clone these genes? This should be spelled out, perhaps as an explicit description in the Materials and methods section. Complete tables of all orthologs attempted for all the donor species tested, along with reasons for failure for the 9 that missed in bacteria for example, should be provided in the supplement.*

To obtain clear loss-of-function phenotypes, we chose all *E. coli* orthologs of essential yeast genes with no lineage specific duplications (i.e., only 1:1 orthologs). Though in total there are 460 shared orthogroups between *E. coli* and yeast (as per InParanoid 8), only 58 fit the criteria of 1) yeast essentiality and 2) 1:1 orthology. We have now clarified this process in the text and in the Materials and methods section, and corrected a typo in the prior Figure 1 vs. 58). As now described in the text (subsection “Many *E. coli* genes successfully complement lethal defects in their yeast orthologs”, first paragraph; subsection “Ortholog inference”), we used Inparanoid 8 to identify the 58 *E. coli* genes that are 1:1 orthologs of essential yeast genes. We cloned and confirmed the sequence of all 58 of these *E. coli* genes in yeast expression vectors. Of these, 51 provided informative assays, 5 were inconclusive, and 2 had no matched yeast strains available to test replaceability. We have modified the [Supplementary-material SD1-data]appropriately.

*2) From what I could conclude, in both bacteria and Arabidopsis there was an issue with complementing Sc-hem12. The bacteria one could not be integrated and both Arabidopsis orthologs failed to complement. What is the issue with Hem12? How come the human ortholog Hs-UROD did complement? Even if there is no single clear answer to these questions a hypothesis would be welcome.*

Our observations were that the bacterial ortholog (*Ec-hemE*) of Sc-*HEM12* functionally replaced only when constitutively expressed under the GPD promoter. Human *Hs-UROD* also replaced when so expressed. However, plant versions did not work when constitutively expressed on plasmids. Thus, to address the issue of why we observed differential replaceability across variants of this gene, we performed the following analyses:

A) In the case of the Ec-*HemE* bacterial replacement at the yeast genomic locus, we suspect that the reason for not obtaining a replacement is that we used only 60bp of sequence homology to the flanking yeast locus, limiting the efficiency of the homologous repair. However, it is still possible that the bacterial version at the yeast native locus does not replace the yeast gene function, explaining the lack of positive clones. We did not pursue this particular case in more depth.

B) The human *Hs-UROD* version available in the human ORFeome had a single mutation resulting in a single amino acid change G303V. This variant is non-replaceable as explained in the text (subsection “Each yeast heme biosynthesis enzyme can be replaced by its human ortholog”, first paragraph). Reverting this mutation back to wild type (encoding glycine) allowed successful replacement of the yeast gene (Figure 6—figure supplement 3).

C) The plant co-orthologs of the yeast gene Sc-*Hem12 (*AtHEME1/E2) did not replace the yeast gene when expressed under the control of the GPD promoter. In response to the referee’s queries, we have now tested and eliminated two possible reasons for this lack of replacement:

i) Plant heme pathway proteins possess chloroplast localization signals (CLS) at their N-termini, and we showed two cases where GFP-tagged plant proteins localize to mitochondria in yeast (At-PPOX1 and At-FC-I). However, the Sc- *Hem12* reactions take place in the cytosol. We therefore first suspected mislocalization to the mitochondria to be the likely reason for non-replaceability. We have now tested whether the removal of the At-HEME1 or AtHEME2 CLS would allow functional replacement; however, this was unsuccessful (Figure 6—figure supplement 1’’’). We also tested At-HEMC, an initially poor replacer. For this gene, removal of the CLS significantly enhanced replaceability compared to the wild type protein (Figure 6—figure supplement 1’’), demonstrating that the CLS did indeed contribute to non-replaceability in some cases.

ii) We next suspected functional divergence or sub-functionalization as a potential contributor to the lack of complementation. We co-expressed both paralogs (testing both the wild type and CLS-less versions) under the control of a GPD promoter on two different plasmids with different selections for transformation (SD-Ura and Hygromycin). Co-expression of both genes in the same strain failed to functionally replace the yeast gene function (Figure 6—figure supplement 1’’’), ruling out sub-functionalization as a likely reason for the failure to complement.

We speculate that there could be several other reasons why complementation failed, including unknown intermediate reactions, required localization in a special compartment (e.g. chloroplast) or different transcriptional/translational regulation in plants that might contribute to the lack of functional replaceability.

We have incorporated the new data into the manuscript, and indicate (subsection “Most yeast heme biosynthesis enzymes can also be successfully plant-ized”, last paragraph) that we tested multiple hypotheses to attempt to explain these trends.

*3) For the hemH pink phenotype, to show that this is actually due to the specific substrate proposed, they should delete Sc-HEM14 (or another upstream function) to genetically verify that the pink phenotype disappears because formation of protoprophyrin IX should be prevented by this "upstream" mutation.*

We have now performed additional experiments to confirm our hypothesis regarding protoporphyrin IX accumulation. Using CRISPR, we deleted the *Sc-HEM14* ORF in wild type BY4741, Sc-*hem15*Δ::Ec-*HemH,* and Sc-*hem15*Δ::Ec-MLS-*HemH* strains. Consistent with protoporphyrin IX being the pink pigment in the Sc-*hem15*Δ::Ec-*HemH* strain, the Sc- *hem15*Δ::Ec-*HemH hem14*Δ strain lost the pink phenotype, even after growing for 6 days.

Moreover, we observed that all strains carrying the *hem14*Δ allele were in fact significantly paler than even wild type BY4741 cells, presumably reflecting extensive protoporphyrin IX depletion in these cells. These data are now provided in Figure 5—figure supplement 2.

Reviewer #2:

*[…] Apart from the general point above, the main interest of this work lies in the analysis of various predictors and correlates of ortholog replaceability. I do not share the authors' surprise regarding the lack of correlation between replaceability and sequence conservation. It has been shown in a number of analyses that there is at best a very limited connection between sequence conservation and gene essentiality (Jordan IK, Rogozin IB, Wolf YI, Koonin EV. Essential genes are more evolutionarily conserved than are nonessential genes in bacteria. Genome Res. 2002 Jun;12(6):962-8; Wang Z, Zhang J. Why is the correlation between gene importance and gene evolutionary rate so weak? PLoS Genet. 2009 Jan;5(*1*):e1000329). I think we see here a manifestation of the same phenomenon: sequence conservation depends much stronger on the abundance of a protein product and the gene-specific functional constraints than on the "importance".*

*The shape of the dependency in Figure 3 seems paradoxical at first glance (non-monotonic curve, with moderately conserved genes being most replaceable) but I suspect is explained by the different in replaceability among functional classes of genes (Figure 3). I find it highly desirable to test this directly and discuss accordingly.*

Though the majority of the proteins tested had moderate sequence conservation, we saw no particular relationship between sequence conservation and functional replaceability. We now expand on this point and have incorporated the citations mentioned by the referee in the subsection “Conclusions” (first paragraph). We additionally tested for the enrichment of particular GO Biological Process categories within each bin of sequence identity from Figure 3. Those genes in the 40-50% category had an enrichment in glucose metabolism (3 of the 7 genes). Other than that bin, no other category had any significant enrichment in biological processes or KEGG pathways. We now discuss this point specifically in the first paragraph of the subsection “Replaceability varies strongly across different biological processes”.

*To me, the results in Figure 3 are indeed the most interesting in the paper. Again, this looks striking and at least superficially, paradoxical, in that genes in the most highly conserved categories, such as translation and tRNA modification, are virtually non-replaceable. I believe the explanation lies in the complexity hypothesis (Jain R, Rivera MC, Lake JA. Horizontal gene transfer among genomes: the complexity hypothesis. Proc Natl Acad Sci U S A. 1999 Mar 30;96(8):3801-6) that seems to be the best explanation for the rates of horizontal gene transfer in different functional classes of prokaryotic genes. Indeed, ortholog replacement studied here can be considered an extreme, "forced" variant of horizontal gene transfer. I think a thoughtful discussion of these parallels and their utility for explaining the results could make the present story considerably more interesting.*

Our results in Figure 3 do seem to agree with the complexity hypothesis, in that housekeeping genes are more likely to be replaceable while informational genes are not. We have added a short discussion of this topic in the subsection “Conclusions” (first paragraph).

*Reviewer #3:*

*This is a fun and thought-provoking study that systematically replaces essential yeast genes with their orthologs from E. coli. There is good rescue by many of the 1:1 orthologs. The authors then more extensively investigate the 'swappability' of all the proteins in the haem biosynthesis pathway, finding that even mislocalised proteins can rescue some yeast deletion phenotypes as can swapping in an alternative (non-orthologous) part of the pathway. This extends their previous work (and the work of two other labs) performing similar experiments with human 1:1 orthologs over a larger evolutionary distance, formally showing that, at least for metabolic enzymes, function has been conserved over huge evolutionary distances to the extent that the enzymes can be swapped from one species to another.*

*Minor suggestions/queries:*

*1. The authors do not describe how the two complementation assays work in the main text or how much agreement there is between them when they are directly compared for the same ORFs. I think they should do both to assist the general reader. The rescue of TS mutants is straightforward, but only yeast aficionados will understand the second assay. In addition, this cannot be at all understood from their sentence in the main text: '…were carried out using two types of conditionally essential yeast alleles, consisting of temperature- sensitive (TS) haploid and heterozygous knockout diploid yeast strains”. In Figure 1 and elsewhere it would help to indicate how many ORFs and which were tested with which complementation assay, how many with both and the agreement between them.*

We now describe the assays in the main text (Results). A more detailed assay description is also listed in the Methods section. For 11 cases, we obtained informative assays for both the TS and MM alleles of the same gene. 10 of these assays shared the same complementation status, whereas 1 did not. We have now updated [Supplementary-material SD1-data] to clearly indicate results from both assays.

2. Are any of the mutant phenotypes not rescued by overexpressing the yeast ORF?

This is of course an important control. We previously tested the general complementation rate of deletion alleles of essential yeast genes by plasmid-based copies of the same genes under the GPD promoter and found they replaced at a rate of 100% in 29 strains tested (Kachroo et al., Science, 2015). As we are using the same strain collections here, we expect a comparable (high) rate. We have now performed additional control experiments to confirm this: First, we tested whether 6 yeast deletion strains, which could not be rescued by their corresponding *E. coli* orthologs, became replaceable if the corresponding yeast genes were similarly heterologously expressed on a CEN plasmid. In all 6 cases, complementation was successful (Figure 1—figure supplement 2 and Sc-HEM1 as reported in Figure 4—figure supplement 1). Second, we specifically tested the entire heme biosynthesis pathway for replaceability by the corresponding yeast genes when expressed either under the control of the heterologous GPD promoter or the native promoter (using MOBY collection yeast ORF plasmids) (Figure 4—figure supplement 1). In all but one case, the yeast genes replaced and the mode of expression was irrelevant to the efficiency of replaceability (Figure 4—figure supplement 1). However, similar to the human ortholog Hs-UROS, the expression of the yeast gene, Sc-HEM4, was toxic when expressed under the control of the constitutive GPD promoter (Figure 4—figure supplement 1 & Figure 6—figure supplement 3). This toxicity was relieved if the yeast protein was expressed under the native promoter, again similar to the human Hs-UROS, which showed functional replacement when integrated at the genomic locus (Figure 4—figure supplement 1 and Figure 6—figure supplement 3). These additional experiments have now been incorporated into the manuscript where indicated above.

Additional changes in the revised version: In addition to the suggestions of the reviewers, we also now plot all growth assays as a mean and standard deviation of N=3 replicate growth curves.